# VD3D: Taming Large Video Diffusion Transformers for 3D Camera Control

**Sherwin Bahmani**[1,2,3]    **Ivan Skorokhodov**[3]    **Aliaksandr Siarohin**[3]    **Willi Menapace**[3]
**Guocheng Qian**[3]    **Michael Vasilkovsky**[3]    **Hsin-Ying Lee**[3]    **Chaoyang Wang**[3]
**Jiaxu Zou**[3]    **Andrea Tagliasacchi**[1,4]    **David B. Lindell**[1,2]    **Sergey Tulyakov**[3]
[1]University of Toronto  [2]Vector Institute  [3]Snap Inc.  [4]SFU

https://snap-research.github.io/vd3d

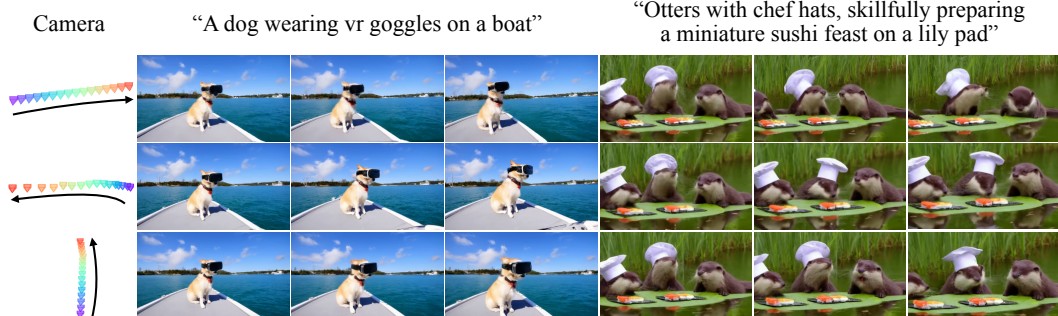

Figure 1: **3D camera control for text-to-video generation.** We introduce a method that can control camera poses for text-to-video generation using video diffusion transformers. (left) The method takes as input a set of camera poses used to generate each frame of a rendered video. (center, right) Applying multiple camera trajectories with the same text prompt enables synthesis of complex scenes from a varied set of viewpoints. Video results: https://snap-research.github.io/vd3d.

## Abstract

Modern text-to-video synthesis models demonstrate coherent, photorealistic generation of complex videos from a text description. However, most existing models lack fine-grained control over camera movement, which is critical for downstream applications related to content creation, visual effects, and 3D vision. Recently, new methods demonstrate the ability to generate videos with controllable camera poses—these techniques leverage pre-trained U-Net-based diffusion models that explicitly disentangle spatial and temporal generation. Still, no existing approach enables camera control for new, transformer-based video diffusion models that process spatial and temporal information jointly. Here, we propose to tame video transformers for 3D camera control using a ControlNet-like conditioning mechanism that incorporates spatiotemporal camera embeddings based on Plücker coordinates. The approach demonstrates state-of-the-art performance for controllable video generation after fine-tuning on the RealEstate10K dataset. To the best of our knowledge, our work is the first to enable camera control for transformer-based video diffusion models.

## 1 Introduction

Text-to-video foundation models achieve unprecedented visual quality (Brooks et al., 2024; Sharma et al., 2024). They are trained on massive collections of images and videos and learn to synthesize remarkably consistent and physically plausible visualizations of the world. Yet, they lack built-in mechanisms for explicit 3D control during the synthesis process, requiring users to manipulate outputs through prompt engineering and trial and error—a slow, laborious, and computationally expensive process. For example, as Fig. 2 shows, state-of-the-art video models struggle to follow even simple "zoom-in" or "zoom-out" camera trajectories using text prompt instructions (see supplemental webpage). This lack of controllability limits interactivity and makes existing video generation

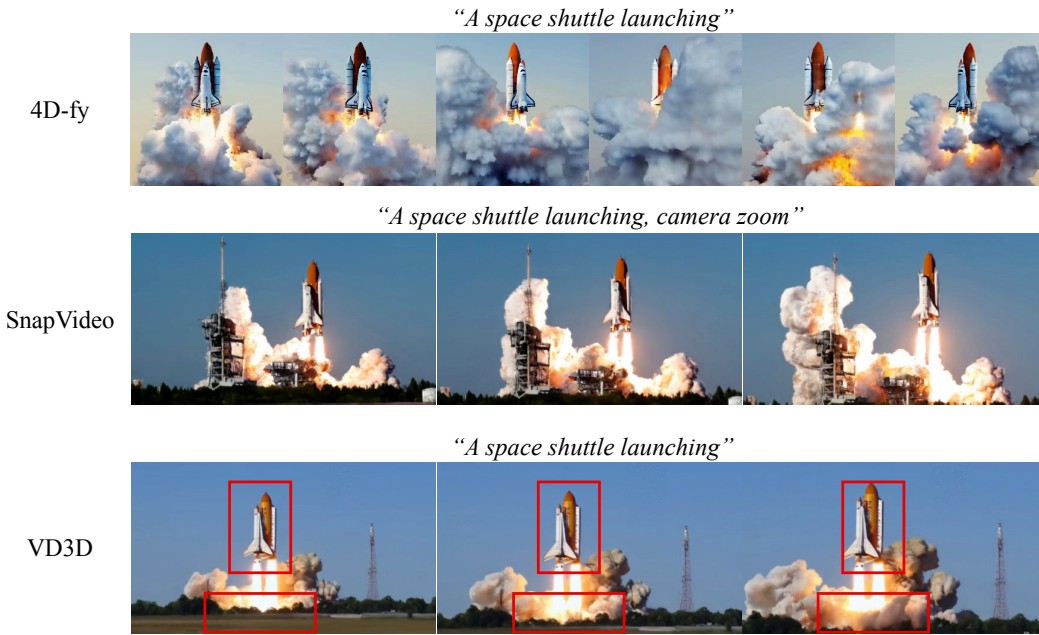

Figure 2: **Comparing text-to-video, text-to-4D, and camera-conditioned text-to-video generation.** We show time progressing across columns for generated videos from different approaches. Text-to-4D approaches, such as 4D-fy (Bahmani et al., 2024b) (top), have complete control over the camera through a 3D representation but lack photorealism. We visualize time progressing while simultaneously changing the viewpoint. (middle) Methods for text-to-video generation (Menapace et al., 2024) create realistic videos but do not provide explicit control over the viewpoint. In contrast, camera-conditioned text-to-video generation (bottom) bridges the gap between the two paradigms by extending text-to-video generators with 3D camera control without using an explicit 3D representation. Specifically, we condition VD3D with a zoom-in camera trajectory to control the generation.

techniques challenging to use for artists or other end users. We augment 2D video generation models with control over the position and orientation of the camera, providing finer-grained control compared to text prompting, and facilitating use of video generation models for downstream applications.

Several contemporary works (Wang et al., 2023e; He et al., 2024a; Yang et al., 2024b) propose methods for camera control of state-of-the-art, open-source video diffusion models. The key technical insight proposed by these methods is to add camera control by fine-tuning the temporal conditioning layers of a U-Net-based video generation model on a dataset with high-quality camera annotations. While these techniques achieve promising results, they are not applicable to more recent, high-quality transformer-based architectures (Vaswani et al., 2017; Peebles & Xie, 2023), such as Sora (Brooks et al., 2024), SnapVideo (Menapace et al., 2024), and Lumina-T2X (Gao et al., 2024a), as these latest works simply do not have standalone temporal layers amenable to camera conditioning.

Large video transformers represent a video as a (possibly compressed) sequence of tokens, applying self-attention layers to all the tokens jointly (Brooks et al., 2024; Menapace et al., 2024). Consequently, as diffusion transformers do not have standalone temporal layers, they are incompatible with current camera conditioning approaches. As the community shifts towards large video transformers to jointly model spatiotemporal dependencies in the data, it is critical to develop methods that provide similar capabilities for camera control. Our work designs a camera conditioning method tailored to the joint spatiotemporal computation used in large video transformers and takes a step towards taming them for controllable video synthesis.

We develop our work on top of our implementation of SnapVideo (Menapace et al., 2024), a state-of-the-art video diffusion model, that employs Far-reaching Interleaved Transformers (FIT) blocks (Chen & Li, 2023) for efficient video modeling in the compressed latent space. We investigate various camera conditioning mechanisms in the fine-tuning scenario and explore trade-offs in terms of visual quality preservation and controllability. Our findings reveal that simply adapting existing approaches

to video transformers does not yield satisfactory results: they either enable some limited amount of control while reducing the visual quality of the output video, or they entirely fail to control camera motion. Our key technical insight is to enable the control through spatiotemporal camera embeddings, which we derive by combining Plücker coordinates with the network input through a separately trained cross-attention layer. To the best of our knowledge, our work is the first to explore a ControlNet-like (Zhang et al., 2023) conditioning mechanism for spatiotemporal transformers.

We evaluate the method on a collection of manually crafted text prompts and unseen camera trajectories and compare to baseline approaches that incorporate previous camera control methods into a video transformer. Our approach achieves state-of-the-art results in terms of camera controllability and video quality, and also enables downstream applications such as multi-view, text-to-video generation, as depicted in Figure 1. In contrast to existing image-to-3D methods (e.g., (Liu et al., 2024a; Qian et al., 2024b; Voleti et al., 2024a)), which are limited to object-centric scenes, our approach synthesizes novel views for real input images with complex environments.

Overall, our work makes the following contributions.

- We propose a new method to tame large video transformers for 3D camera control. Our approach uses a ControlNet-like conditioning mechanism that incorporates spatiotemporal camera embeddings based on Plücker coordinates.

- We thoroughly evaluate this approach, including comparisons to previous camera control methods, which we adapt to the video transformer architecture.

- We show state-of-the-art results in camera-controllable video synthesis by applying the proposed conditioning method and fine-tuning scheme to the SnapVideo-based model (Menapace et al., 2024).

## 2 RELATED WORK

Our method is connected to techniques related to text-to-video, text-to-3D, and text-to-4D generation. As this is a popular and fast-moving field, this section provides only a partial overview with a focus on the most relevant techniques; we refer readers to Po et al. (2023) and Yunus et al. (2024) for a more thorough review of related techniques.

**Text-to-video generation**. Our work builds on recent developments in 2D video generation models. One such class of these techniques works by augmenting text-to-image models with layers that operate on the temporal dimension to facilitate video generation (Blattmann et al., 2023b; Singer et al., 2023a; Wu et al., 2023; Guo et al., 2024; Blattmann et al., 2023a). Video generation models can be trained in a hybrid fashion on both images and videos to improve the generation quality (Bain et al., 2021; Wang et al., 2023b; Xue et al., 2022; Ho et al., 2022a; Guo et al., 2024; He et al., 2022; Wang et al., 2023c; Zhou et al., 2022). While they are primarily based on convolutional, U-Net-style architectures, a recent shift towards transformer-based architectures enables synthesis of much longer videos with significantly higher quality (Brooks et al., 2024; Ma et al., 2024b; Menapace et al., 2024; Ma et al., 2024a). Still, these methods do not enable synthesis with controllable camera motion.

**4D generation**. Previous methods also tackle the problem of 4D generation, i.e., generating videos of dynamic 3D scenes from controllable viewpoints, usually from an input text prompt or image. Since the initial work on this topic using large-scale generative models (Singer et al., 2023b), significant improvements in the visual quality and motion quality of generated scenes have been achieved (Ren et al., 2023; Ling et al., 2024a; Bahmani et al., 2024b; Zheng et al., 2024a; Bahmani et al., 2024a). While these methods generate scenes based on text conditioning, other approaches convert an input image or video to a dynamic 3D scene (Ren et al., 2023; Zhao et al., 2023; Yin et al., 2023; Pan et al., 2024; Zheng et al., 2024a; Ling et al., 2024a; Gao et al., 2024b; Zeng et al., 2024; Chu et al., 2024). Another line of work (Bahmani et al., 2023a; Xu et al., 2023) extends 3D GANs into 4D by training on 2D videos, however the quality is limited and models are trained on single category datasets. All of these methods are focused on object-centric generation, typically based on 3D volumetric representations. As such, they typically do not incorporate background elements, and overall, they do not approach the level of photorealism demonstrated by the state-of-the-art video generation models used in our technique (see Fig. 2).

**Controllable generation with diffusion models**. Methods for controllable generation using diffusion models have had significant impact, both in the context of image and video generation. For example, existing techniques allow controllable image generation conditioned on text, depth maps, edges, pose, or other signals (Zhang et al., 2023; Ye et al., 2023). While Chen et al. (2024) developed a ControlNet-based mechanism for transformer-based diffusion, limited to spatial conditioning with spatial signals, we explore conditioning mechanisms for camera poses (a spatio-temporal signal with an intricate temporal component) in a spatio-temporal transformer. Furthermore, there is a line of work for 3D generation that conditions diffusion models on camera poses for view-consistent multi-view generation (Watson et al., 2023; Tseng et al., 2023; Chan et al., 2023; Yu et al., 2023a; Kumari et al., 2024; Müller et al., 2024; Gao et al., 2024c). Our approach is most similar to related techniques in video generation that seek to control the camera position. For example, MotionCtrl (Wang et al., 2023e) designs camera and object control mechanisms for the VideoCrafter1 (Chen et al., 2023b) and SVD (Blattmann et al., 2023a) models. However, MotionCtrl is designed for U-Net-based approaches and does not directly apply to video diffusion transformers.

**Concurrent 3D camera control methods**. Concurrent approaches enable camera control by conditioning the temporal layers of the network with camera pose information, e.g., using Plücker coordinates (He et al., 2024a; Guo et al., 2024; Xu et al., 2024b; Kuang et al., 2024) or other embeddings (Yang et al., 2024b). Interestingly, it is also possible to incorporate camera control into video generation models without additional training through manipulation and masking of attention layers, though this requires additional tracking, segmentation, or depth for each input video (Hu et al., 2024; Xiao et al., 2024; Hou et al., 2024). Another recent work (Ling et al., 2024b) transfers motion, including camera motion, to other generated videos.

Although these approaches demonstrate promising results for U-Net-based video diffusion models, the techniques are not applicable to modern video transformers that model spatio-temporal dynamics jointly. While another concurrent work (Watson et al., 2024) uses a transformer-based architecture for space and time, it does not tackle text-based generation for dynamic scenes but focuses on novel view synthesis from an input image. In our work, we design an efficient mechanism that enables camera control in video diffusion transformers using a ControlNet inspired mechanism without sacrificing visual quality.

## 3 METHOD

### 3.1 LARGE TEXT-TO-VIDEO TRANSFORMERS

**Text-to-video generation**. Diffusion models have emerged as the dominant paradigm for large-scale video generation (Ho et al., 2022b;a; Brooks et al., 2024). The standard setup considers the conditional distribution $p(\boldsymbol{x}|\boldsymbol{y})$ of videos $\boldsymbol{x} \in \mathbb{R}^{F \times H \times W}$ (consisting of $F$ frames of $H \times W$ resolution) given their text descriptions $\boldsymbol{y} \in \mathcal{Y}^L$, consisting of $L$ (possibly padded) tokens from the alphabet $\mathcal{Y}$. Following (Karras et al., 2022), our video diffusion framework assumes a denoising model $D_{\boldsymbol{\theta}} : (\tilde{\boldsymbol{x}}; \boldsymbol{y}, \sigma) \mapsto \hat{\boldsymbol{x}}$ that predicts a clean video $\hat{\boldsymbol{x}}$ from the corresponding noised input $\tilde{\boldsymbol{x}} = \boldsymbol{x} + \sigma\boldsymbol{\varepsilon}$, where $\boldsymbol{\varepsilon} \sim \mathcal{N}(\boldsymbol{0}, \boldsymbol{I})$ is standard Gaussian noise and $\sigma \sim \log\mathcal{N}(P_{\text{mean}}, P_{\text{std}}^2)$ is the noise strength, sampled from log-normal distribution with location $P_{\text{mean}}$ and scale $P_{\text{std}}$. The model is parametrized by a neural network $F_{\boldsymbol{\theta}}(\tilde{\boldsymbol{x}}; \boldsymbol{y}, \sigma)$ as $D_{\boldsymbol{\theta}}(\tilde{\boldsymbol{x}}; \boldsymbol{y}, \sigma) = c_{\text{out}}(\sigma)F_{\boldsymbol{\theta}}(c_{\text{in}}(\sigma)\tilde{\boldsymbol{x}}; \boldsymbol{y}, \sigma) + c_{\text{skip}}(\sigma)\tilde{\boldsymbol{x}}$, where $c_{\text{in}}(\sigma), c_{\text{out}}(\sigma)$ and $c_{\text{skip}}(\sigma)$ are input, output and residual scaling factors from (Karras et al., 2022). The minimization objective is defined as

$$\mathcal{L}(\boldsymbol{\theta}) = \mathbb{E}_{p(\boldsymbol{x},\boldsymbol{y},\sigma,\boldsymbol{\varepsilon})} \left[ \|D_{\boldsymbol{\theta}}(\boldsymbol{x} + \sigma\boldsymbol{\varepsilon}; \boldsymbol{y}, \sigma) - \boldsymbol{x}\|_2^2 \right]. \tag{1}$$

We refer the reader to (Menapace et al., 2024) and (Karras et al., 2022) for further details on the diffusion setup, which we adopted without modifications.

**Spatiotemporal transformers**. Following SnapVideo (Menapace et al., 2024), our video generator consists of two models: the base 4B-parameters generator, operating on 16-frames $36 \times 64$ resolution videos, and a $288 \times 512$ upsampler. The latter, a diffusion model itself, is fine-tuned from the base model and conditioned on the generated low-resolution videos. Each model uses FIT transformer blocks (Chen & Li, 2023; Jabri et al., 2022) for efficient self-attention operations (see Fig. 3). An FIT model consists of $B$ blocks (we have $B = 6$ in all the experiments) and first partitions each frame in an input video into patches (Dosovitskiy et al., 2021) of resolution $h_p \times w_p$ (we use

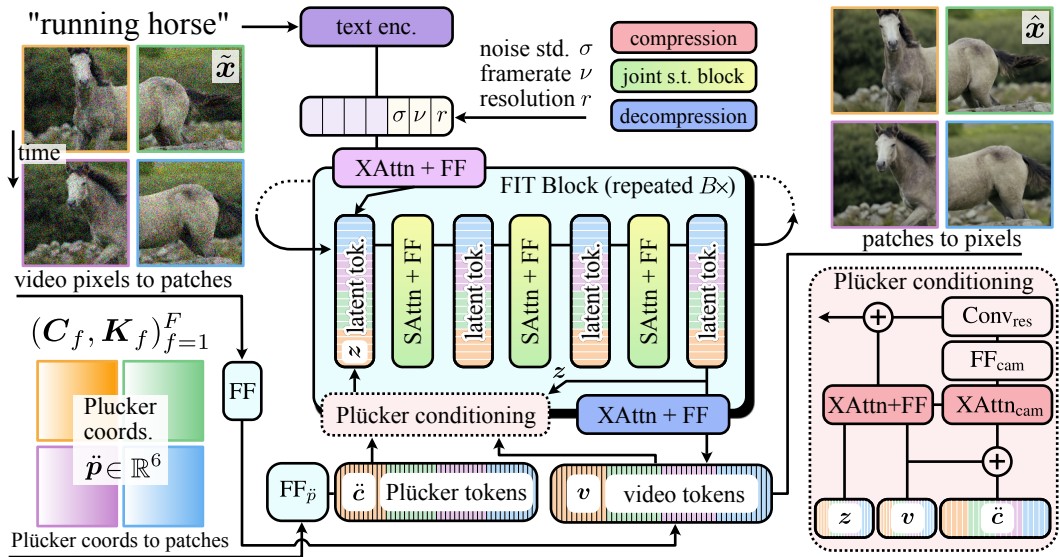

Figure 3: **Overview of architecture.** We adapt a FIT-based architecture (Chen & Li, 2023) to incorporate camera control. We take as input the noisy input video $\tilde{x}$, camera extrinsics $C_f$, and camera intrinsics $K_f$ for each video frame $f$. We compute the Plücker coordinates for each pixel within the video frames using the camera parameters. Both the input video and Plücker coordinate frames are converted to patch tokens, and we condition the video patch tokens using a mechanism similar to ControlNet (Zhang et al., 2023) ("Plücker conditioning" block). Then, the model estimates the denoised video $\hat{x}$ by recurrent application of FIT blocks (Chen & Li, 2023). Each block reads information from the patch tokens into a small set of latent tokens on which computation is performed. The results are written to the patch tokens in an iterative denoising diffusion process.

$h_p = w_p = 4$ in all the experiments). These video patches are then independently projected via a feedforward (FF) layer to obtain a sequence of video tokens $v_{\ell:L}^{(b)} \triangleq (v_1, ..., v_L) \in \mathbb{R}^{L \times d}$ of length $L = F \times (H/h_p) \times (W/w_p)$ and dimensionality $d$. Next, each FIT block "reads" the information from this video sequence into a much shorter sequence of $M$ *latent* tokens $z_{m:M}^{(b)} \triangleq (z_1, ..., z_M)$ through a "read" cross-attention layer, followed by a feedforward layer. The core processing with self-attention layers is performed in this latent space, and then the result is written back to the video tokens through a corresponding "write" cross-attention layer (also followed by an FF layer). The latent tokens in each subsequent FIT block are initialized from the previous one, which helps to propagate the computational results throughout the network. In this way, the entire computation occurs jointly in both spatial and temporal axes, which yields superior scalability (Menapace et al., 2024). However, it abandons the decomposed spatial/temporal computation nature of modern video diffusion U-Nets, which modern camera conditioning techniques (Wang et al., 2023e; Yang et al., 2024b) rely on to enforce control without compromising visual quality.

## 3.2 CAMERA CONTROL FOR SPATIOTEMPORAL TRANSFORMERS

**Spatiotemporal camera representation**. The standard way of representing camera parameters for a video (in the pinhole model) is via a trajectory of extrinsics and intrinsics camera parameters $(C_f, K_f)_{f=1}^F$ for each $f$-th frame. The matrix $C_f = [R; t] \in \mathbb{R}^{3 \times 4}$, describes the camera rotation $R \in \mathbb{R}^{3 \times 3}$ and translation $t \times \mathbb{R}^3$, and $K_f \in \mathbb{R}^{3 \times 3}$ contains the focal length and principal point (and also horizontal/vertical skew coefficient, but it is always 0 in our setup). To control camera motion, existing methods condition the temporal attention layers of U-Net-based video generators on embeddings computed from these camera parameters (Wang et al., 2023e; Yang et al., 2024b; Hu et al., 2024). Such a pipeline provides a good conditioning signal for convolutional video generators with decomposed spatial/temporal computation, but our experiments demonstrate that it works poorly for spatiotemporal transformers: they either fail to pick up any controllability (when being added as transformed residuals to the latent tokens), or degrade the visual quality of the output (when

the original network parameters are being fine-tuned). This motivates us to design a better camera conditioning scheme, tailored for modern large-scale spatiotemporal transformers.

First, we propose to normalize the camera parameters w.r.t the first frame. For this, we recompute the rotations and translations for each $f$-th frame as $\boldsymbol{R}'_f = \boldsymbol{R}_1^{-1}\boldsymbol{R}_f$ and $\boldsymbol{t}'_f = \boldsymbol{t}_f - \boldsymbol{t}_1$. This procedure results in normalized camera extrinsics as $\boldsymbol{C_f}' = [\boldsymbol{R}'_f; \boldsymbol{t}'_f]$ and establishes a consistent coordinate system across different samples in the dataset. After that, we found it essential to enrich the conditioning information by switching from temporal frame-level camera parameters to pixel-wise *spatiotemporal* ones. This is achieved by computing the Plücker coordinates for each pixel, providing a fine-grained positional representation.

Plücker coordinates provide a convenient parametrization of lines in the 3D space, and we use them to compute fine-grained positional representations of each pixel in each frame of a video. Given the extrinsic and intrinsic camera parameters $\boldsymbol{R}'_f, \boldsymbol{t}'_f, \boldsymbol{K}_f$ of the $f$-th frame, we parametrize each $(h, w)$-th pixel as a Plücker embedding $\ddot{\boldsymbol{p}}_{f,h,w} \in \mathbb{R}^6$ from the camera position to the pixel's center as

$$\ddot{\boldsymbol{p}}_{f,h,w} = (\boldsymbol{t}_f' \times \hat{\boldsymbol{d}}_{f,h,w}, \hat{\boldsymbol{d}}_{f,h,w}), \quad \hat{\boldsymbol{d}}_{f,h,w} = \frac{\boldsymbol{d}}{\|\boldsymbol{d}_{f,h,w}\|}, \quad \boldsymbol{d}_{f,h,w} = \boldsymbol{R}_f'\boldsymbol{K}_f[w, h, 1]^\top + \boldsymbol{t}_f'. \quad (2)$$

This approach mirrors the technique used in recent 3D works (Sitzmann et al., 2021; Chen et al., 2023a; Kant et al., 2024) as well as CameraCtrl (He et al., 2024a), a concurrent study focusing on camera control in U-Net-based video diffusion models. The motivation for using Plücker coordinates is that geometric manipulations in the Plücker space can be performed through simple arithmetics on the coordinates, which makes it easier for the network to use the positional information stored in such a disentangled representation.

Computing Plücker coordinates for each pixel results in a $\ddot{\boldsymbol{P}} \in \mathbb{R}^{6 \times F \times H \times W}$ spatiotemporal camera representation for a video. To input it into the model, we first perform the equivalent ViT-like (Dosovitskiy et al., 2021) $h_p \times w_p$ patchification procedure. It is followed by a learnable 2-layered MLP $\text{MLP}_{\ddot{p}}$ with a GELU (Hendrycks & Gimpel, 2016) non-linearity to obtain the Plücker camera tokens sequence $\ddot{\boldsymbol{c}}_{\ell:L} \in \mathbb{R}^{L \times d}$ of the same length $L = F \times (H/h_p) \times (W/w_p)$ and dimensionality $d$ as the video tokens sequence $\boldsymbol{v}_{\ell:L}^{(b)}$. This spatiotemporal representation carries fine-grained positional information about each pixel in a video, making it easier for the generator to accurately follow the desired camera motion.

**Camera conditioning**. To input the Plücker embeddings into our video generator, we design an efficient ControlNet like (Zhang et al., 2023) mechanism tailored for large transformer models (see Fig. 3). This mechanism is guided by two main objectives: 1) the model should be amenable to rapid fine-tuning from a small dataset with estimated camera positions; and 2) the visual quality shouldn't be compromised during the fine-tuning stage. We found that meeting these objectives is more challenging for spatiotemporal transformers compared to U-Net-based models with decomposed spatial/temporal computation, since even minor interventions into their design quickly lead to degraded video outputs. We hypothesize that the core reason for it is the entangled spatial/temporal computation of video transformers: any attempt to alter the temporal dynamics (such as camera motion) influences spatial communication between the tokens, leading to unnecessary signal propagation and overfitting during the fine-tuning stage. To mitigate this, we input the camera information gradually through `read` cross-attention layers, zero-initialized from the original network parameters of the corresponding layers.

Specifically, in each $b$-th FIT block of our video generator, we replace its standard `read` cross-attention operation (see (Jabri et al., 2022; Menapace et al., 2024) for details):

$$\boldsymbol{z}'^{(b)}_{m:M} = \text{FF}^{(b)}(\text{XAttn}^{(b)}(\boldsymbol{z}^{(b)}_{m:M}, \boldsymbol{v}^{(b)}_{\ell:L})), \quad (3)$$

where $\text{FF}(\cdot)$ and $\text{XAttn}(\cdot, \cdot)$ denote feed-forward and cross-attention layers respectively. Our revised formulation is given as:

$$\boldsymbol{z}'^{(b)}_{m:M} = \text{FF}^{(b)}(\text{XAttn}^{(b)}(\boldsymbol{z}^{(b)}_{m:M}, \boldsymbol{v}^{(b)}_{\ell:L})) + \text{Conv}^{(b)}_{\text{res}}\left[\text{FF}^{(b)}_{\text{cam}}(\text{XAttn}^{(b)}_{\text{cam}}(\boldsymbol{z}^{(b)}_{m:M}, \ddot{\boldsymbol{c}}_{\ell:L} + \boldsymbol{v}^{(b)}_{\ell:L}))\right], \quad (4)$$

where $\text{FF}^{(b)}_{\text{cam}}$, and $\text{XAttn}^{(b)}_{\text{cam}}$ are learnable layers, $\text{Conv}^{(b)}_{\text{res}}$ is a 1-dimensional convolution that processes camera-augmented latents. The produced latents $\boldsymbol{z}'^{(b)}_{m:M}$ are then passed to the sequence of four

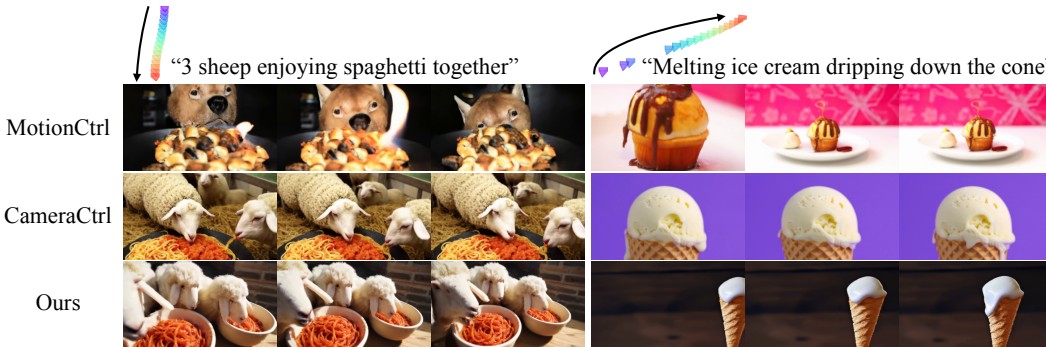

Figure 4: **Camera-conditioned text-to-video generation.** Comparison of the proposed approach to MotionCtrl and CameraCtrl for the same camera trajectory input. We adapted MotionCtrl and CameraCtrl to the same transformer-based model for fair comparisons. MotionCtrl exhibits worse quality due to fine-tuning existing layers, and CameraCtrl is not sensitive to camera conditioning. See the supplementary webpage for video results.

self-attention layers, which are the core computational component of FIT. It is crucial to instantiate the weights of the output convolutions $\text{Conv}_{\text{res}}^{(b)}$ from zeros to preserve the model initialization. Besides, we initialize the weights of $\text{FF}_{\text{cam}}^{(b)}$ and $\text{XAttn}_{\text{cam}}^{(b)}$ from the corresponding parameters of the original network. This approach helps to preserve visual quality at initialization and facilitates rapid fine-tuning on a small dataset. As a result, we obtain the method for fine-grained 3D camera control in large video diffusion transformers. We name it VD3D and visualize its architecture in Figure 3.

### 3.3 TRAINING DETAILS

To ensure comparability with prior work such as (Wang et al., 2023e), we train our video generator on the same RealEstate10K dataset (Zhou et al., 2018). We optimize only the newly added parameters $\text{FF}_{\ddot{p}}$ and $(\text{FF}^{(b)}, \text{XAttn}^{(b)})_{b=1}^{B}$, and keep the rest of the network frozen. We found that training only the base $36 \times 64$ model was sufficient, as the $288 \times 512$ upsampler already accurately follows the camera motion of a low-resolution video.

### 3.4 DATASET

We fine-tune a text-to-video model, pre-trained on 2D video data, on RealEstate10K (Zhou et al., 2018). The training split for fine-tuning consists of roughly 65K video clips, and is the same as is used in concurrent work (MotionCtrl (Wang et al., 2023e) and CameraCtrl (He et al., 2024a)).

### 3.5 METRICS

We conduct a user study to evaluate our approach for camera-controlled text-to-video generation. The study participants are presented with 20 side-by-side comparisons between the proposed approach and the baselines as well as a reference video from RealEstate10K with the same trajectory to better judge the camera alignment. We ask 20 participants for each generated video sequence to indicate which generated video they prefer based on multiple submetrics, namely, camera alignment, motion quality, text alignment, visual quality, and overall preference. The user study involved negligible risk to the participants and was conducted with appropriate institutional review board and legal approval. We evaluate our method using 20 camera trajectories sampled from the RealEstate10K test split that were not seen during training for the user study. We use the full test split combined with unseen text prompts for the automated camera evaluations, i.e., 6928 unseen camera trajectories combined with out-of-distribution text prompts.

### 3.6 BASELINES

We compare our work to MotionCtrl (Wang et al., 2023e) and the concurrent work CameraCtrl (He et al., 2024a) by adapting their publicly released code to the same pre-trained video model as ours. Note that both baselines were originally designed for space-time disentangled U-Net video diffusion models. Hence, their approaches are not directly applicable to the spatio-temporal transformers, and so we adapt them to this setting as follows. For MotionCtrl, we omit their object motion control module and use their proposed camera motion control module to encode the camera parameters into a context vector used with the SnapVideo model. We fine-tune both the camera motion control module and the cross attention between the latent vectors and the patches. For CameraCtrl, we fine-tune the original camera encoder module and use this to produce the latent vectors in the SnapVideo model. During fine-tuning the model weights are kept frozen—i.e., the same as in our proposed approach. Furthermore, we include comparisons to the original MotionCtrl and CameraCtrl works built on the U-Net-based AnimateDiff (Guo et al., 2024) architecture. For fair comparison, we evaluate all metrics for MotionCtrl (U-Net) and CameraCtrl (U-Net) on the same test sets as our method using the publicly provided code and checkpoints. Moreover, we provide metrics for the base model used across all our experiments, i.e., a model without camera injection. We provide more baseline variants of MotionCtrl and CameraCtrl in the appendix in Sec. B.3.

## 4 EXPERIMENTS

### 4.1 ASSESSMENT

We provide a qualitative and quantitative assessment of our approach compared to the baselines in Fig. 4 and in Tab. 1. Following CameraCtrl (He et al., 2024a), we also evaluate the camera pose accuracy using ParticleSfM (Zhao et al., 2022) on generated videos in Tab. 2. We use generations for text prompts from RealEstate10K (Zhou et al., 2018) and MSR-VTT (Xu et al., 2016), testing both in- and out-of-distribution prompts. Note that we adjust the CameraCtrl (He et al., 2024a) evaluation pipeline by normalizing all cameras into a unified scale as COLMAP provides different scales across different scenes. This prevents scenes with large scale to have a higher impact on the errors. Please also refer to the supplementary webpage for additional video results.

In Fig. 4 we observe that adapting the camera conditioning method from the MotionCtrl degrades visual quality and text alignment, likely because this approach adjusts the weights of the base video model. In the space-time U-Net for which this approach was proposed, the temporal layers can be fine-tuned without sacrificing visual fidelity. Since spatio-temporal transformers do not decompose temporal and spatial attributes in the same way, the model overfits to the small dataset used to fine-tune the cross-attention layer. While we observe some agreement with the camera poses used to condition the model, the text alignment is generally low in our experiments (see supplemental webpage). In contrast, CameraCtrl keeps the pre-trained video model weights frozen and only trains a camera encoder. This leads to strong visual quality, but the generated videos show little agreement with the input camera poses. For fair comparison, we trained all models for the same number of iterations (described in Sec. 3.3).

The results of the user study in Tab. 1 show that most participants prefer the generated videos using the proposed camera conditioning mechanism across all evaluated sub-metrics. We also observe a pronounced preference for the camera alignment of the proposed method compared to the other baselines. That is, 82% and 78% of participants prefer the camera alignment of the proposed method compared to our respective adaptations of MotionCtrl and CameraCtrl to the video transformer model. All results are significant at the $p < 0.001$ level as evaluated using a $\chi^2$ test. We further present image-based metrics for multi-view generation and video generation quality metrics in the appendix in Sec. B.1 and Tab. B.2 respectively. The results of our camera pose accuracy evaluation in Tab. 2 further demonstrate that our approach clearly outperforms previous works. Note that the previous works MotionCtrl and CameraCtrl especially struggle with the rotation accuracy.

### 4.2 ABLATIONS

**Plücker embedding.** We motivate our Plücker embedding conditioning mechanism by training a variant using the raw camera matrices. Concretely, we flatten and concatenate extrinsics and intrinsics

Table 1: **Quantitative results.** We compare our method to MotionCtrl and CameraCtrl implemented on the same base video models as ours. The methods are evaluated in a user study in which participants indicate their preference based on camera alignment (CA), motion quality (MQ), text alignment (TA), visual quality (VQ), and overall preference (Overall). The percentages indicate preference for VD3D vs. the alternative method (in each row). All results are statistically significant with $p < 0.001$ as evaluated using a $\chi^2$ test.

| | Human Preference | | | | |
| Method | CA | MQ | TA | VQ | Overall |
|---|---|---|---|---|---|
| VD3D vs. MotionCtrl | 82% | 81% | 86% | 81% | 84% |
| VD3D vs. CameraCtrl | 78% | 64% | 63% | 65% | 66% |

Table 2: **Camera pose evaluation.** We evaluate all models using reference camera trajectories from the RealEstate10K test set. We compute translation and rotation errors based on estimated camera poses from generations using ParticleSfM (Zhao et al., 2022).

| Method | RealEstate10K | | MSR-VTT | |
|---|---|---|---|---|
| | TransError ($\downarrow$) | RotError ($\downarrow$) | TransError ($\downarrow$) | RotError ($\downarrow$) |
| Base Model | 0.616 | 0.207 | 0.717 | 0.216 |
| MotionCtrl (U-Net) | 0.477 | 0.094 | 0.593 | 0.137 |
| CameraCtrl (U-Net) | 0.465 | 0.089 | 0.587 | 0.132 |
| MotionCtrl | 0.518 | 0.161 | 0.627 | 0.148 |
| CameraCtrl | 0.532 | 0.165 | 0.578 | 0.220 |
| Ours | **0.409** | **0.043** | **0.504** | **0.050** |
| w/o Plucker | 0.517 | 0.161 | 0.676 | 0.156 |
| w/o ControlNet | 0.573 | 0.182 | 0.787 | 0.179 |
| w/o weight copy | 0.424 | 0.044 | 0.513 | 0.063 |
| w/o add context | 0.602 | 0.212 | 0.702 | 0.128 |
| w/o Plucker context | 0.487 | 0.088 | 0.627 | 0.091 |

matrices in the channel dimension and repeat the values in the spatial patch dimensions. We observe that Plücker embeddings provide an essential spatial conditioning mechanism, as shown in Tab. 2.

**ControlNet conditioning.** Our ControlNet-inspired conditioning mechanism ensures fast and precise learning of the conditioning signal distribution. Instead of using a ControlNet block, we simply add zero-initialized Plücker embedding features to the patches and observe close to no camera control. We observe training cross-attention layers in the ControlNet block is key to learning camera control while preserving the original model weights. This is confirmed by our camera evaluation in Tab. 2.

**ControlNet weight copy.** While the ControlNet block is essential, copying the pre-trained weights into the new copy has rather minor impact, as shown in Tab. 2. To verify this, we train a model where we randomly initialize the cross-attention block between patches and latents instead of copying the weights. We observe similar results, showing that rather the architecture and zero-initialization are the key component of this design.

**Add to context vector.** We train a model where we use camera matrices, linearly encode them, and add them to the context vector as a simple conditioning mechanism for transformer-based models. The rendered videos show little to no correlation with the input camera matrices, resulting in poor camera pose accuracy, as shown in Tab. 2. This highlights the importance of carefully incorporating cameras into the spatio-temporal patches with our Plücker embedding block.

**Plücker in context vector.** We train a model where we integrate Plücker embeddings into the context vector instead of the patches. We similarly compute Plücker embeddings, but instead of patchifying them, we flatten and linearly map the embeddings into a vector that has the same shape as the context vector. We fuse the Plücker features and context vector using the same Plücker conditioning mechanism used for the patches. We observe higher camera pose errors in Tab. 2, highlighting that it

Camera      "A bedroom with a bed, lamps and a window"      "A house sitting in the middle of a grassy field"

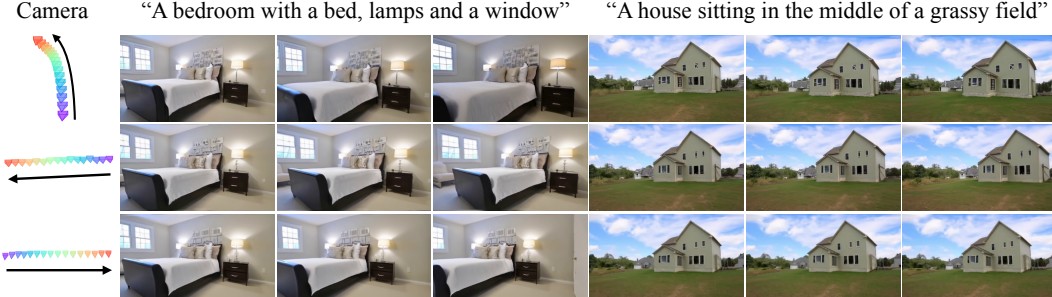

Figure 5: **Conditional multi-view generation on a real image.** We can generate arbitrary camera trajectories from a given real image for multi-view synthesis, paving the way to single-image scene reconstruction using camera-controlled video models.

is crucial to incorporate the spatio-temporal Plücker embeddings into the spatio-temporal patches instead of using the context vector.

### 4.3 APPLICATIONS

**Image-to-video generation.** We synthesize camera-controlled videos based on different camera poses as shown in Fig. 1. For this task we use a version of our pre-trained text-to-video model that we fine-tune on video sequences where a random subset of the input frames are masked. At inference time, we can provide image guidance for any of the generated frames, providing an additional dimension of controllability when paired with our proposed method for camera control. To demonstrate image-to-video generation in Fig. 1, we condition the model using camera poses along with image guidance from the first frame of a generated video sequence. Note that our method provides no control over motion within the scene itself; hence, scene motion can differ depending on the random seed or the provided camera poses.

**Image-to-multiview generation.** We also explore multi-view generation for static scenes as shown in Fig. 5. Given a real input image of a complex scene unseen during training, our camera-conditioned model generates view-consistent renderings of that scene from arbitrary viewpoints. These multi-view renderings could be directly relevant to downstream 3D reconstructions pipelines, e.g., based on NeRF (Mildenhall et al., 2020) or 3D Gaussian Splatting (Kerbl et al., 2023). We show the potential of camera-conditioned image-to-multiview generation for complex 3D scene generation, but we leave more extensive exploration of this topic for future work.

## 5 CONCLUSION

Large-scale video transformer models show immense promise to solve many long-standing challenges in computer vision, including novel-view synthesis, single-image 3D reconstruction, and text-conditioned scene synthesis. Our work brings additional controllability to these models, enabling a user to specify the camera poses from which video frames are rendered.

**Limitations and future work.** There are several limitations to our work, which highlight important future research directions. For example, while rendering static scenes from different camera viewpoints produces results that appear 3D consistent, dynamic scenes rendered from different camera viewpoints can have inconsistent motion (see supplemental videos). We envision that future video generation models will have fine-grained control over both scene motion and camera motion to address this issue. Further, our approach applies camera conditioning only to the low-resolution SnapVideo model and we keep their upsampler model frozen (i.e., without camera conditioning)—it may be possible to further improve camera control through joint training, though this brings additional architectural engineering and computational challenges. Finally, our approach is currently limited to generation of relatively short videos (16 frames), based on the design and training scheme of the SnapVideo model. Future work to address these limitations will enable new capabilities for applications in computer vision, visual effects, augmented and virtual reality, and beyond.

## 6 ETHICS STATEMENT

**Broader Impact.** Recent video generation models demonstrate coherent, photorealistic synthesis of complex scenes—capabilities that are highly sought after for numerous applications across computer vision, graphics, and beyond. Our key technical contributions relate to camera control of these models, which can be applied to a wide range of methods. As with all generative models and technologies, underlying technologies can be misused by bad actors in unintended ways. While these methods continue to improve, researchers and developers should continue to consider safeguards, such as output filtering, watermarking, access control, and others.

**Data.** To develop the camera control methods proposed in this paper, we used RealEstate10K (Zhou et al., 2018). RealEstate10K is released and open-sourced by Google LLC under a Creative Commons Attribution 4.0 International License, and sourced from content using a CC-BY license. The dataset can be found under the following URL: `https://google.github.io/realestate10k`. The RealEstate10K dataset was published as part of a research paper by Zhou et al. (2018): "Stereo Magnification: Learning View Synthesis using Multiplane Images" at SIGGRAPH 2018. The RealEstate10K dataset is a commonly used dataset for 3D reconstruction (Charatan et al., 2024), 3D generation (Gao et al., 2024c), and camera-controlled video generation (Wang et al., 2023e). Similarly, closely related baselines such as MotionCtrl (Wang et al., 2023e) and CameraCtrl (He et al., 2024a) use the identical dataset to train and evaluate their camera-controlled video diffusion models. Hence, we follow an established training and evaluation pipeline without any modifications to the data.

## 7 REPRODUCIBILITY STATEMENT

We have structured our paper to ensure comprehensive reproducibility of our camera control method. Section 3 provides a detailed description of our method, including theoretical foundations and core algorithmic components. Section 4 presents thorough experimental details, covering evaluation protocols, metrics, and comparisons with baselines. All implementation specifics, including training procedures, architectural choices, and hyperparameters, are documented in Appendix C. To further facilitate reproducibility, we include the source code of our camera-controlled FIT block as supplementary material. The provided implementation contains the core components necessary to replicate our approach, accompanied by documentation and usage examples. We welcome requests for additional technical details or clarifications to support the research community in reproducing and building upon our work.

## 8 ACKNOWLEDGEMENTS

DBL acknowledges support from NSERC under the RGPIN program, the Canada Foundation for Innovation, and the Ontario Research Fund.

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

## A  RELATED WORK

Due to space constraints, we summarize related 3D and an extended list of 4D works in the appendix.

**3D generation.**  Limited to single category trainings, early work on 3D generation extends GANs into 3D using a neural renderer as an inductive bias (DeVries et al., 2021; Chan et al., 2022; Or-El et al., 2022; Schwarz et al., 2022; Bahmani et al., 2023b), Towards more diverse and flexible generation, CLIP-based supervision (Radford et al., 2021) enabled text-based generation and editing of 3D assets (Chen et al., 2018; Jain et al., 2022; Sanghi et al., 2022; Jetchev, 2021; Gao et al., 2023; Wang et al., 2022). With recent advances in diffusion models, Score Distillation Sampling (SDS) replaces CLIP supervision with diffusion model supervision (Poole et al., 2023; Wang et al., 2023d; Lin et al., 2023a; Chen et al., 2023c; Liang et al., 2023; Wang et al., 2023a; Li et al., 2024d; He et al., 2024b; Ye et al., 2024; Liu et al., 2024c; Yu et al., 2023b; Katzir et al., 2024; Lee et al., 2024a; Sun et al., 2024a) for higher quality generation. In order to improve the 3D structure of scenes, another line of work generates multiple views of a scene (Lin et al., 2023b; Liu et al., 2023; Shi et al., 2024; Feng et al., 2024a; Liu et al., 2024b; Kim et al., 2023; Voleti et al., 2024b; Höllein et al., 2024). Alternatively, other methods use iterative inpainting for 3D scene generation (Höllein et al., 2023; Shriram et al., 2024). Recent methods lift input images into 3D (Chan et al., 2023; Tang et al., 2023; Gu et al., 2023; Liu et al., 2024d; Yoo et al., 2023; Tewari et al., 2023; Qian et al., 2024c; Long et al., 2024; Wan et al., 2024; Szymanowicz et al., 2023) using NeRF (Mildenhall et al., 2020), 3D Gaussian Splatting (Kerbl et al., 2023), or Meshes in combination with diffusion models. Other recent methods (Hong et al., 2024; Li et al., 2024b; Xu et al., 2024d;c; Zhang et al., 2024a; Han et al., 2024; Jiang & Wang, 2024; Xie et al., 2024; Tang et al., 2024; Tochilkin et al., 2024; Qian et al., 2024a; Szymanowicz et al., 2024b;a) tackle fast feed-forward 3D generation, directly predicting a 3D generation from input images or text. Different from our approach, these methods can only synthesize static scenes.

**4D generation**. Previous methods also tackle the problem of 4D generation, i.e., generating videos of dynamic 3D scenes from controllable viewpoints, usually from an input text prompt or image. Since the initial work on this topic using large-scale generative models (Singer et al., 2023b), significant improvements in the visual quality and motion quality of generated scenes have been achieved (Ren et al., 2023; Ling et al., 2024a; Bahmani et al., 2024b; Zheng et al., 2024a; Gao et al., 2024b; Yang et al., 2024a; Jiang et al., 2024; Bahmani et al., 2024a; Zhang et al., 2024d; Xu et al., 2024a; Miao et al., 2024; Li et al., 2024a). While these methods generate scenes based on text conditioning, other approaches convert an input image or video to a dynamic 3D scene (Ren et al., 2023; Zhao et al., 2023; Yin et al., 2023; Pan et al., 2024; Zheng et al., 2024a; Ling et al., 2024a; Gao et al., 2024b; Zeng et al., 2024; Chu et al., 2024; Wu et al., 2024; Yang et al., 2024c; Wang et al., 2024; Feng et al., 2024b; Sun et al., 2024b; Zhang et al., 2024b; Yu et al., 2024; Ren et al., 2024; Lee et al., 2024b; Li et al., 2024c; Van Hoorick et al., 2024; Uzolas et al., 2024; Huang et al., 2024; Chai et al., 2024; Liang et al., 2024; Zhang et al., 2024c). Another line of work (Bahmani et al., 2023a; Xu et al., 2023) extends 3D GANs into 4D by training on 2D videos, however the quality is limited and models are trained on single category datasets. Still, all of these methods are focused on object-centric generation, typically based on 3D volumetric representations. As such, they typically do not incorporate background elements, and overall, they do not approach the level of photorealism demonstrated by the state-of-the-art video generation models used in our technique.

## B  QUANTITATIVE EVALUATION

We further evaluate all models for the task of single image-to-multiview generation. Furthermore, we provide results for established 2D video generation metrics.

### B.1  MULTI-VIEW GENERATION

We evaluate our model for image-to-multiview generation. Due to ground truth correspondences for the RealEstate10K (Zhou et al., 2018) dataset, we can condition our model on a given camera trajectory and assess image based metrics, i.e., PSNR, SSIM, and LPIPS. We provide results for the low-resolution base model and the upsampled high-resolution results in Tab. 3.

## B.2 QUALITY METRICS

Moreover, we evaluate all models on established image and video generation metrics, namely, FID (Heusel et al., 2017), FVD (Unterthiner et al., 2018), and CLIPSIM (Wu et al., 2021). We provide results for RealEstate10K (Zhou et al., 2018) and MSR-VTT (Xu et al., 2016) in Tab. 4.

It is important to highlight that FID and FVD are not direct measurements of visual quality—these metrics measure similarities between dataset distributions. So while fine-tuning on a dataset will degrade these scores relative to the original model, this degradation is expected because the model is fitting a different data distribution than the one used in the original training run. The same degradation in FID after fine-tuning is observed in the original ControlNet paper (Zhang et al., 2023), where the authors report an increase in FID when comparing the StableDiffusion base model (6.09 FID) to their fine-tuned ControlNet model (15.27 FID; see their Table 3). We believe that joint training with 2D video data may help to alleviate this degradation, and we will explore joint training strategies as part of future work.

## B.3 BASELINE VARIANTS

We provide further variants of MotionCtrl and CameraCtrl adopted to the SnapVideo base model. These variants performed worse than the ones presented in the main paper, hence we mainly show them here for completeness to show the different implementations we explored.

On top of that, we train a variant of MotionCtrl where instead of integrating the camera matrices into the context vector, we integrate them into the output of the cross-attention layer between the latent tokens and patches (as in our approach). We observe degraded camera control compared to our approach. Moreover, we train a variant of CameraCtrl where we incorporate the Plucker embeddings of the CameraCtrl camera encoder into the patches instead of the latents. We observe worse camera control compared to our model. Furthermore, we train a variant of MotionCtrl where we freeze all base model layers similar to CameraCtrl and our approach. We use the same batch size, same number of iterations, and same parameter size as our proposed method for fair comparisons. Note that MotionCtrl unfreezes the attention layer after injecting camera features into the base model. This experiment highlights that we outperform MotionCtrl independent of training or freezing these attention layers. We show results for camera pose accuracy, multi-view generation, and video quality in Tab. 5, Tab. 6, and Tab. 7 respectively.

## B.4 GENERALIZATION OF CAMERA TRAJECTORIES

We conduct additional experiments with horizontal and vertical panning, where the camera trajectory is defined by rotation-only camera matrices without any translation. Specifically, we manually construct each trajectory by randomly selecting either the x-, y-, or z-axis and randomly sampling the angular extent of the trajectory (from 0 to 120 degrees). We set the camera translation to the zero vector. We show results in Tab. 8 and observe higher accuracy for our methods. Furthermore, we include results for non-random, user-defined camera trajectories that involve camera movements with significant directional changes including both rotations and translations in Tab. 9. These include following trajectories: rotation around clockwise; rotation around anticlockwise; rotation clockwise without translation; rotation anticlockwise without translation; zoom out, then up translation; translation right, then rotation anticlockwise; translation left, then rotation clockwise; translation left; translation right; translation up; translation down. We observe that our method generalizes to input camera trajectories with variable rotations and translations.

## B.5 EXPERIMENTS WITH VANILLA DIT

Instead of building upon the FIT (Chen & Li, 2023) architecture, we implemented our VD3D method on top of a pre-trained text-to-video DiT model (Peebles & Xie, 2023) in the latent space of the CogVideoX (Yang et al., 2024d) autoencoder. We include these results in Tab. 10 and Tab. 11. Instead of building on top of read attention in FIT, we incorporate the ControlNet conditioning on top of the vanilla attention mechanism of the actual tokens. We observe that the vanilla DiT version further improves quality and camera accuracy on out-of-distribution prompts (MSR-VTT). We believe that

our proposed method of spatio-temporal Plucker tokens and aligning them with video patch tokens through a ControlNet-type of conditioning mechanism is agnostic to the transformer architecture and will serve as a starting point for follow-up works.

**Base video DiT architecture details.** The video DiT architecture follows the design of other contemporary video DiT models (e.g., Sora (Brooks et al., 2024), MovieGen (Polyak et al., 2024), OpenSora (Zheng et al., 2024b), LuminaT2X (Gao et al., 2024a), and CogVideoX (Yang et al., 2024d)). As the backbone, it incorporates a transformer-based architecture with 32 DiT blocks. Each DiT block includes a cross-attention layer for processing text embeddings (produced by the T5-11B model), a self-attention layer, and a fully connected network with a ×4 dimensionality expansion. Attention layers consist of 32 heads with RMSNorm for query and key normalization. Positional information is encoded using 3D RoPE attention, where the temporal, vertical, and horizontal axes are allocated fixed dimensionality within each attention head (using a 2:1:1 ratio). LayerNorm is applied to normalize activations within each DiT block. A pre-trained CogVideoX autoencoder is utilized for video dimensionality reduction, employing causal 3D convolution with a 4×8×8 compression rate and 16 channels per latent token. The model features a hidden dimensionality of 4,096 and comprises 11.5B parameters. It leverages block modulations to condition the video backbone on rectified flow timestep information, SiLU activations, and 2×2 ViT-like patchification of input latents to reduce sequence size.

**Base video DiT training details.** The base DiT model is optimized using AdamW, with a learning rate of 0.0001 and weight decay of 0.01. It is trained for 750,000 iterations with a cosine learning rate scheduler in bfloat16. Image animation support is incorporated by encoding the first frame with the CogVideoX encoder, adding random Gaussian noise (sampled independently from the video noise levels), projecting via a separate learnable ViT-like patchification layer, repeating sequence-wise to match video length, and summing with the video tokens. Training incorporates loss normalization and is conducted jointly on images and videos with variable resolutions (256, 512, and 1024), aspect ratios (16:9 and 9:16 for videos; 16:9, 9:16, and 1:1 for images), and video lengths (ranging from 17 to 385 frames). Videos are generated at 24 frames per second, and variable-FPS training is avoided due to observed performance decreases for target framerates without fine-tuning.

**Base video DiT inference details.** Inference uses standard rectified flow without stochasticity. We find forty steps to balance quality and sampling speed effectively. For higher resolutions and longer video generation, a time-shifting strategy similar to Lumina-T2X is used, with a time shift of $\sqrt{32}$ for 1024-resolution videos.

## C    TRAINING DETAILS

**Compute details**. A single training run for the smaller 700M parameter generator takes approximately 1 day on a node equipped with $8\times$ NVIDIA A100 40GB GPUs, connected via NVIDIA NVLink, along with 960 GB of RAM and 92 Intel Xeon CPUs. The larger 4B parameter model was trained on 8 such nodes for 1,5 days, totaling $64\times$ NVIDIA A100 40GB GPUs. In total, we conducted $\approx$150 training runs for the smaller model during the development stage of 4B generator. Consequently, the project's total compute utilization amounted to approximately 2700 NVIDIA A100 40GB GPU-days.

**Optimization details**. We experiment with two model variants: a smaller generator with approximately 700 million parameters for ablations and initial explorations, and a larger 4 billion parameter model, which we use for the main results in this paper. Both models were trained with a batch size of 256 over 50,000 optimization steps with the LAMB optimizer (You et al., 2019). The learning rate was warmed up for the first 10,000 iterations from 0 to 0.005 and then linearly decreased to 0.0015 over subsequent iterations. For the large model, VD3D contains 230M trainable parameters in total which corresponds to around 5% of the total amount. Since the original video diffusion model is trained in the any-frame-conditioning pipeline (Menapace et al., 2024), we can produce variable camera trajectories from the same starting frame. For text conditioning, we use the T5-11B (Raffel et al., 2020) language model to encode text 1024-dimensional embeddings into 128-length sequences. For training efficiency, they were precomputed for the entire dataset. The rest of the training details have been adopted from (Menapace et al., 2024).

Table 3: **Multi-view generation.** We evaluate all models using reference camera trajectories and single-view input images of the RealEstate10K test set. We compute reconstruction metrics based on the subsequent frames for the low-resolution and upsampled high-resolution generations.

| Method | Low-resolution | | | High-resolution | | |
|---|---|---|---|---|---|---|
| | PSNR ($\uparrow$) | SSIM ($\uparrow$) | LPIPS ($\downarrow$) | PSNR ($\uparrow$) | SSIM ($\uparrow$) | LPIPS ($\downarrow$) |
| Base Model | 14.74 | 0.320 | 0.334 | 13.23 | 0.459 | 0.572 |
| MotionCtrl (U-Net) | 15.35 | 0.387 | 0.294 | 13.64 | 0.473 | 0.548 |
| CameraCtrl (U-Net) | 15.86 | 0.412 | 0.266 | 13.81 | 0.479 | 0.540 |
| MotionCtrl | 15.07 | 0.348 | 0.308 | 13.42 | 0.467 | 0.560 |
| CameraCtrl | 14.81 | 0.327 | 0.330 | 13.21 | 0.456 | 0.571 |
| Ours | **17.23** | **0.534** | **0.211** | **14.90** | **0.499** | **0.499** |
| w/o Plucker | 14.89 | 0.346 | 0.308 | 13.05 | 0.455 | 0.573 |
| w/o ControlNet | 14.66 | 0.313 | 0.340 | 13.10 | 0.450 | 0.573 |
| w/o weight copy | 16.96 | 0.509 | 0.220 | 14.75 | 0.495 | 0.504 |
| w/o add context | 14.45 | 0.303 | 0.368 | 13.06 | 0.446 | 0.588 |
| w/o Plucker context | 14.76 | 0.322 | 0.318 | 13.28 | 0.463 | 0.579 |

Table 4: **Quality metrics evaluation.** We evaluate all models using text prompts from the RealEstate10K and MSR-VTT test sets respectively.

| Method | RealEstate10K | | | MSR-VTT | | |
|---|---|---|---|---|---|---|
| | FID ($\downarrow$) | FVD ($\downarrow$) | CLIPSIM ($\uparrow$) | FID ($\downarrow$) | FVD ($\downarrow$) | CLIPSIM ($\uparrow$) |
| Base Model | 8.22 | 160.37 | 0.2677 | **3.50** | **141.26** | **0.2774** |
| MotionCtrl (U-Net) | 2.99 | 61.70 | 0.2646 | 16.85 | 283.12 | 0.2411 |
| CameraCtrl (U-Net) | 2.48 | 55.64 | 0.2681 | 12.33 | 201.33 | 0.2505 |
| MotionCtrl | 1.50 | 52.30 | 0.2708 | 9.97 | 183.57 | 0.2677 |
| CameraCtrl | 2.28 | 66.31 | 0.2730 | 8.47 | 181.90 | 0.2690 |
| Ours | 1.40 | 42.43 | **0.2807** | 7.80 | 165.18 | 0.2689 |
| w/o Plucker | **1.17** | 43.65 | 0.2715 | 9.84 | 152.91 | 0.2660 |
| w/o ControlNet | 3.66 | 137.06 | 0.2766 | 8.34 | 185.79 | 0.2674 |
| w/o weight copy | 1.38 | **42.00** | 0.2710 | 10.09 | 218.43 | 0.2647 |
| w/o add context | 1.45 | 44.74 | 0.2735 | 9.56 | 173.43 | 0.2657 |
| w/o Plucker context | 1.54 | 43.88 | 0.2724 | 8.83 | 168.95 | 0.2615 |

Table 5: **Camera pose evaluation with additional baselines.** We evaluate additional variants of the baselines using reference camera trajectories from the RealEstate10K test set. We compute translation and rotation errors based on estimated camera poses from generations using ParticleSfM (Zhao et al., 2022).

| Method | RealEstate10K | | MSR-VTT | |
|---|---|---|---|---|
| | TransError ($\downarrow$) | RotError ($\downarrow$) | TransError ($\downarrow$) | RotError ($\downarrow$) |
| MotionCtrl (latents) | 0.549 | 0.183 | 0.650 | 0.145 |
| CameraCtrl (patches) | 0.587 | 0.197 | 0.648 | 0.233 |
| MotionCtrl (frozen) | 0.607 | 0.205 | 0.678 | 0.122 |
| Ours | **0.409** | **0.043** | **0.504** | **0.050** |

Table 6: **Multi-view generation.** We evaluate additional variants of the baselines using reference camera trajectories and single-view input images of the RealEstate10K test set. We compute reconstruction metrics based on the subsequent frames for the low-resolution and upsampled high-resolution generations.

| Method | Low-resolution | | | High-resolution | | |
|---|---|---|---|---|---|---|
| | PSNR (↑) | SSIM (↑) | LPIPS (↓) | PSNR (↑) | SSIM (↑) | LPIPS (↓) |
| MotionCtrl (latents) | 14.67 | 0.323 | 0.331 | 13.23 | 0.467 | 0.562 |
| CameraCtrl (patches) | 14.42 | 0.304 | 0.366 | 13.04 | 0.450 | 0.577 |
| MotionCtrl (frozen) | 14.59 | 0.308 | 0.340 | 13.11 | 0.455 | 0.573 |
| Ours | **17.23** | **0.534** | **0.211** | **14.90** | **0.499** | **0.499** |

Table 7: **Quality metrics evaluation with additional baselines.** We evaluate additional variants of the baselines using text prompts from the RealEstate10K and MSR-VTT test sets respectively.

| Method | RealEstate10K | | | MSR-VTT | | |
|---|---|---|---|---|---|---|
| | FID (↓) | FVD (↓) | CLIPSIM (↑) | FID (↓) | FVD (↓) | CLIPSIM (↑) |
| MotionCtrl (latents) | 1.83 | 77.39 | 0.2788 | 10.21 | 187.42 | 0.2636 |
| CameraCtrl (patches) | 2.57 | 71.04 | 0.2703 | 9.84 | 184.22 | 0.2612 |
| MotionCtrl (frozen) | 3.53 | 142.15 | 0.2772 | 8.19 | 165.48 | 0.2679 |
| Ours | **1.40** | **42.43** | **0.2807** | **7.80** | **165.18** | **0.2689** |

Table 8: **Camera pose evaluation for random rotation trajectories.** We evaluate all models using trajectories with randomly selected x-, y-, or z-axis and randomly sampled angular extent of the trajectory. We compute translation and rotation errors based on estimated camera poses from generations using ParticleSfM (Zhao et al., 2022).

| Method | RealEstate10K | | MSR-VTT | |
|---|---|---|---|---|
| | TransError (↓) | RotError (↓) | TransError (↓) | RotError (↓) |
| MotionCtrl | 0.396 | 0.087 | 0.417 | 0.120 |
| CameraCtrl | 0.381 | 0.092 | 0.433 | 0.138 |
| Ours | **0.202** | **0.037** | **0.265** | **0.044** |

Table 9: **Camera pose evaluation for random trajectories with translations and rotations.** We evaluate all models using 11 trajectories for 1000 prompts with trajectories that involve significant directional changes including both rotations and translations. We compute translation and rotation errors based on estimated camera poses from generations using ParticleSfM (Zhao et al., 2022).

| Method | RealEstate10K | | MSR-VTT | |
|---|---|---|---|---|
| | TransError (↓) | RotError (↓) | TransError (↓) | RotError (↓) |
| MotionCtrl | 0.451 | 0.095 | 0.456 | 0.146 |
| CameraCtrl | 0.369 | 0.088 | 0.479 | 0.135 |
| Ours | **0.236** | **0.041** | **0.258** | **0.050** |

Table 10: **Camera pose evaluation with vanilla video DiT backbone.** We incorporate our approach into a pre-trained vanilla DiT model in the latent space of CogVideoX (Yang et al., 2024d). We compute translation and rotation errors based on estimated camera poses from generations using ParticleSfM (Zhao et al., 2022).

| Method | RealEstate10K | | MSR-VTT | |
|---|---|---|---|---|
| | TransError ($\downarrow$) | RotError ($\downarrow$) | TransError ($\downarrow$) | RotError ($\downarrow$) |
| MotionCtrl | 0.501 | 0.145 | 0.602 | 0.152 |
| CameraCtrl | 0.513 | 0.138 | 0.559 | 0.195 |
| Ours | **0.421** | **0.056** | **0.486** | **0.047** |

Table 11: **Quality metrics evaluation with vanilla video DiT backbone.** We incorporate our approach into a pre-trained vanilla DiT model in the latent space of CogVideoX (Yang et al., 2024d). We evaluate additional variants of the baselines using text prompts from the RealEstate10K and MSR-VTT test sets respectively.

| Method | RealEstate10K | | | MSR-VTT | | |
|---|---|---|---|---|---|---|
| | FID ($\downarrow$) | FVD ($\downarrow$) | CLIPSIM ($\uparrow$) | FID ($\downarrow$) | FVD ($\downarrow$) | CLIPSIM ($\uparrow$) |
| MotionCtrl | 1.37 | 44.62 | 0.2752 | 9.68 | 157.90 | 0.2684 |
| CameraCtrl | 2.13 | 53.72 | 0.2748 | 8.32 | 152.88 | 0.2723 |
| Ours | **1.21** | **38.57** | **0.2834** | **6.88** | **137.62** | **0.2790** |

