# OpenReview forum: "VD3D: Taming Large Video Diffusion Transformers for 3D Camera Control"
_ICLR.cc/2025/Conference — ICLR 2025 Poster_

### Official Review · Reviewer_ZZhZ · 2024-10-30

**Soundness:** 3
**Presentation:** 3
**Contribution:** 3
**Rating:** 6
**Confidence:** 4

**Summary:**

This paper presents a novel approach to incorporating camera movement control into Transformer-based video diffusion models. While camera control has been extensively studied in UNet-based video diffusion models, this area remains largely uncharted for Transformer architectures. The author introduces an additional ControlNet-like network to inject tokenized Plücker embeddings, demonstrating that this design enhances both visual quality and controllability.

**Strengths:**

1. The authors made an important step forward in camera control for the Transformer-based video diffusion model, which is unexplored by previous research;
2. The visual quality and motion dynamics of the generated videos are excellent;
3. The methodology is clearly explained and easy to understand.

**Weaknesses:**

1. The main components of the method have been validated by previous works, for instance, the Plucker embedding as camera representation (CameraCtrl), and training on RealEstate10K(MotionCtrl, CameraCtrl);

2. As mentioned in 1., I would say the specified designs for Transformer architecture are the major contribution of the paper. However, SnapVideo's architecture is not a typical DiT (it has the "read" operation, and the attention is not performed on the actual tokens). It's not clear how to extend the proposed method for a standard video DiT and how the performance would be.

3. For the MotionCtrl baseline, the visual quality degrades when the base model is fine-tuned. Would it be better to freeze the backbone?

4. Would be better to also provide the trainable parameter scale of the two baselines and the proposed method.

**Questions:**

1. Line 272, R_1, t_1 can be interpreted as the original extrinsic or the extrinsic after normalization. It would be better to use R'_1, and t'_1 to represent the extrinsic after normalization;

2. I am not quite clear about the baseline implementation details. From my understanding, for MotionCtrl, one additional learned token is concatenated to the video patch tokens, while for CameraCtrl, the camera encoder produces additional latent tokens. Is that correct?

3. While previous works sacrifice motion dynamics when training on RealEstate10K (mostly static scenes), the video in the supplementary material exhibits better and larger motions. What would be the possible reasons for the difference?

**Details Of Ethics Concerns:**

None.

---

> ### Author Response · Authors · 2024-11-25
>
> We deeply appreciate the reviewer's detailed and positive assessment of our work. Below, we would like to clarify several concerns.
>
> > **Novelty concern (part 1/2): Plucker camera embeddings have already been explored in CameraCtrl.**
>
> We respectfully want to emphasize that our work not only developed the use of Plücker coordinates for camera control independently, but our core technical innovation lies in the transformer-centric design. We developed a new projection scheme that effectively integrates camera information into the representation space of video tokens, which is crucial for joint spatio-temporal processing in transformer architectures. This technical challenge required novel solutions distinct from those used in traditional UNet-based approaches, as transformers handle information flow and processing in fundamentally different ways.
>
> > **Novelty concern (part 2/2): Using RealEstate10K has already been explored in MotionCtrl.**
>
> We believe there is a misunderstanding: we do not argue that the use of  RealEstate10K is a novel component of our work. The key novelty of our paper lies in exploring and designing a lightweight and precise camera conditioning mechanism for transformer-based diffusion models.
>
> > **How would the method perform on top of DiT?**
>
> That’s a great question! We incorporate our approach into a vanilla DiT model trained in the latent space of CogVideoX [1]. We include these results as part of the revised paper in Table 10 and Table 11. We furthermore include visual samples into the *rebuttal.html*.
>
> Instead of building on top of read attention in FIT, we incorporate the ControlNet conditioning on top of the vanilla attention mechanism of the actual tokens. We provide evaluations in the two Tables below and observe that the vanilla DiT version further improves quality and camera accuracy on out-of-distribution prompts (MSR-VTT). We believe that our proposed method of spatio-temporal Plucker tokens and aligning them with video patch tokens through a ControlNet-type of conditioning mechanism is agnostic to the transformer architecture and will serve as a starting point for follow-up works.
>
> |Method|TransError (RE10K)|RotError (RE10K)|TransError (MSR-VTT)|RotError (MSR-VTT)|
> |:--------|:--------:|:--------:|:--------:|:--------:|
> |MotionCtrl|0.518|0.161|0.627|0.148|
> |CameraCtrl|0.532|0.165|0.578|0.220|
> |Ours (FIT DiT)|**0.409**|**0.043**|0.504|0.050|
> |Ours (vanilla DiT)|0.421|0.056|**0.486**|**0.047**|
>
> |Method|FID (RE10K)|FVD (RE10K)|CLIP (RE10K)|FID (MSR-VTT)|FVD (MSR-VTT)|CLIP (MSR-VTT)|
> |:--------|:--------:|:--------:|:--------:|:--------:|:--------:|:--------:|
> |MotionCtrl|1.50|52.30|0.2708|9.97|183.57|0.2677|
> |CameraCtrl|2.28|66.31|0.2730|8.47|181.90|0.2690|
> |Ours (FIT DiT)|1.40|42.43|0.2807|7.80|165.18|0.2689|
> |Ours (vanilla DiT)|**1.21**|**38.57**|**0.2834**|**6.88**|**137.62**|**0.2790**|
>
> [1] Yang et al., Cogvideox: Text-to-video diffusion models with an expert transformer, arXiv 2024
>
> > **MotionCtrl backbones should be frozen.**
>
> We already trained and evaluated a variant of MotionCtrl where the backbone is frozen in Sec. B.3 with comparisons in Tables 5, 6, and 7, as part of the appendix. While the quality and motion is less degraded, this variant further degrades the camera control since such design lacks enough expressivity to follow a viewpoint conditioning signal.
>
> > **The amount of trainable parameters is unknown.**
>
> We use 230M trainable parameters for VD3D. For fair comparisons, we used for both MotionCtrl and CameraCtrl experiments the **identical** number of parameters to also match GPU memory usage. We train all variants with the same batch size and same number of iterations. We included this information in Appendix C.
>
> > **Use R’_f and t’_f to represent normalized extrinsics.**
>
> We agree with the reviewer on this observation and thank them for the valuable suggestion. We updated Section 3 to accommodate this improvement.
>
> > **Baseline implementation details about MotionCtrl/CameraCtrl.**
>
> Correct, MotionCtrl uses a simple MLP to map from input cameras to camera embeddings that are concatenated to the video patch tokens. CameraCtrl uses a more complex camera encoder including temporal attention and 2D convolutions to map from input cameras to camera embeddings.
>
> > **What is the key ingredient of motion preservation?**
>
> The key to preserving motion lies in our carefully designed conditioning mechanism. Through extensive experimentation, we found that two components are crucial: our Plücker-based projection scheme to obtain camera tokens and the way we integrate them into video tokens of a diffusion transformer via cross-attention followed by zero-initialized convolution to preserve the model initialization. This approach effectively balances the challenging trade-off between camera control precision and motion quality. While enforcing camera control typically degrades scene dynamics, our solution achieves precise camera control while preserving natural motion.

---

### Official Review · Reviewer_DiqV · 2024-11-01

**Soundness:** 3
**Presentation:** 3
**Contribution:** 3
**Rating:** 8
**Confidence:** 3

**Summary:**

The paper proposes a first method to condition spatio-temporal transformer-based video diffusion models on camera poses (previous methods focused on pre-trained U-Net-based diffusion models). To this end, they propose a ControlNet-like block that conditions the model on camera embeddings that are based on Plucker coordinates. The paper evaluates the choices made and demonstrates good results, both qualitatively and quantitatively.

**Strengths:**

The paper presents a nice idea on to enable spatio-temporal transformer-based video diffusion models with camera control. It is the first to do so by using a ControlNet-like block in combination with Plucker coordinates, and presents a nice contribution. I value the ablation studies in the paper and the reasoning as to how they arrived at this particular architecture. The paper is quite well-written overall. It's easy enough to follow and appreciate the contribution.

**Weaknesses:**

While I'm positive overall, there are a few weaknesses.

A criticism that could be made is that this paper is a bit of an A+B paper, where A is the ControlNet block and B is the Plucker coordinates. I don't think that's a useful criticism, because at the end of the day it's a sensible thing to do and the authors demonstrate that you need to do a few things to make it work (Table 2).

The paper reads well overall, however, I'm not a fan of Figure 3. It's hard to interpret what goes where, and how this relates to the formulas. I'd suggest to clearly delineate what are the video patch tokens vs. the plucker patch tokens, maybe add the variables from the formulas to the appropriate boxes, and overall structure the figure so that the separate blocks are clearly separate.

Finally, in terms of results, I would have liked to see more examples of the same scenes with different camera control (there are only three examples). Furthermore, most examples use input camera trajectories from scenes that are completely unrelated to the target scene. It be nice to see some that are related -- makes it easier to judge if the generated camera path is good or not.

**Questions:**

Nothing crucial.

---

> ### Author Response · Authors · 2024-11-25
>
> We deeply appreciate the reviewer's detailed and positive assessment of our work. Below, we would like to clarify several concerns.
>
> > **This paper is a bit of an A+B paper.**
>
> We respectfully disagree with the characterization of our work as a simple combination of existing methods. As demonstrated both quantitatively (Table 2) and qualitatively (supplementary materials), merely combining transformer-based video diffusion with standard conditioning approaches yields poor results and imprecise camera control. Developing an effective camera conditioning scheme for spatiotemporal transformers required extensive experimentation to achieve three critical goals: (1) successfully incorporating camera information into the video token representation space, (2) enforcing this naturally weak conditioning signal during synthesis, and (3) maintaining scene motion and visual quality throughout the process.
>
> > **Figure 3 needs to be improved.**
>
> We appreciate this valuable suggestion. We've made several improvements to both Figure 3 and the notation in Section 3. First, we simplified the notation used to describe the method, and then we restructured the diagram to create a clearer connection between its visual elements and the variables used in the method description, making it more intuitive to read. Finally, we improved the layout to make the figure more visually appealing.
>
> > **Evaluation on diverse camera trajectories and different trajectories starting from the same scene.**
>
> We appreciate these valuable suggestions regarding trajectory evaluation. While our original submission included quantitative results for out-of-distribution camera trajectories in Table 8, we have now substantially expanded this analysis. The new evaluation tests 11 distinct out-of-distribution trajectories across 1000 different scenes for both RealEstate10K and MSR-VTT, shown in the Table below. We visualize 16 prompts for these trajectories, resulting in 176 new visual results in total in the *rebuttal.html* as part of the supplementary file, showing additional examples where multiple trajectories are used for the same scene. This comprehensive analysis is now available in Table 9 of the revised manuscript.
>
> |Method|TransError (RE10K)|RotError (RE10K)|TransError (MSR-VTT)|RotError (MSR-VTT)|
> |:--------|:--------:|:--------:|:--------:|:--------:|
> |MotionCtrl|0.451|0.095|0.456|0.146|
> |CameraCtrl|0.369|0.088|0.479|0.135|
> |Ours|**0.236**|**0.041**|**0.258**|**0.050**|

---

> > ### Comment · Reviewer_DiqV · 2024-11-27
> >
> > Thank you for your detailed reply to my comments. Figure 3 is indeed improved though still a bit dense. I quite liked seeing the additional examples in rebuttal.html. It demonstrates that the conditioning works well -- even on OOD trajectories. I'm happy with the rebuttal.

---

> > > ### Comment · Reviewer_uN7u · 2024-11-28
> > > **Request for clarification on contribution**
> > >
> > > Dear Reviewer DiqV,
> > >
> > > Thanks for the detailed review and for sharing your thoughts. I noticed that you also find this paper to be of A+B style. However, you mentioned that the paper presents a nice contribution, which I didn't follow. Can you please clarify on what exactly is the technical contribution of the paper?
> > >
> > > My confusion comes from my impression that the paper is merely putting CameraCtrl together with SnapVideo. Using Plucker coordinates is not new, and using controlnet for transformers is not new. I understand that `the authors demonstrate that you need to do a few things to make it work (Table 2)`, but I found their re-implementation of CameraCtrl into SnapVideo weird (L381). I don't think the authors implemented their CameraCtrl properly by using the correct module copy or zero-convolutions for the camera encoders.
> > >
> > > Please feel free to share your thoughts.

---

> > > > ### Author Response · Authors · 2024-11-28
> > > >
> > > > Dear Reviewer uN7u,
> > > >
> > > > We appreciate your interest in engaging with Reviewer DiqV's assessment of our work. However, as the authors, we would like to address the comparison concerns directly, which we also outlined in our response to Reviewer uN7u.
> > > >
> > > > > **CameraCtrl implementation**
> > > >
> > > > We believe there is a misunderstanding regarding the implementation of CameraCtrl. We do **not** fine-tune the pre-trained camera encoder from AnimateDiff since this would not generalize well, and even zero-convolutions might not easily adapt to the SnapVideo model. What we mean by “original camera encoder module” is the code and the architecture that we adapt into the same base model codebase we use, i.e., we do not use the weights from AnimateDiff-trained camera encoders. We start fine-tuning the camera encoder module from the **same** SnapVideo model, using the **same** batch size, **same** number of iterations, and **same** parameter size as our proposed method for fair comparisons. We will update the draft to emphasize this as part of the next revision of the paper.
> > > >
> > > > > **Novelty**
> > > >
> > > > Furthermore, we highlighted the novelty of our work in our response to Reviewer uN7u. We represent extrinsics and intrinsics as spatio-temporal Plucker embeddings, patchify them into spatio-temporal Plucker tokens, and then align them with the video tokens. This procedure is unique to transformer-based models and was not investigated by any other work, as previous works only investigated U-Net-based video models. Second, we propose to align these spatiotemporal Plucker tokens with the video tokens using a ControlNet-inspired setup. Other works, including MotionCtrl and CameraCtrl, do not use ControlNet to infuse cameras into the base video model. We observe that spatio-temporals ControlNets are very effective and degrade little motion and quality, as shown in the experiments. Note that using ControlNet is only possible after representing cameras in the same spatio-temporal patch space as the original video tokens as part of the first step we are doing.
> > > > Novelty is not always about inventing a new complicated layer or mechanism, but it can also be novel connecting different mechanisms and showing that a simple approach is more effective. In fact, the community typically prefers to build upon simple approaches that just work. We are not claiming to invent Plucker or ControlNet; it’s the technical solution of representing cameras as spatiotemporal Plucker tokens, using a ControlNet-type of setup to align them spatiotemporal video tokens pixel-wise, and demonstrating that this technique is significantly more effective than any other proposed technique. We also demonstrate with extensive experiments that using raw extrinsics (MotionCtrl) or camera encoders without ControlNet-type of camera alignment (CameraCtrl) work significantly worse, degrading quality, motion, and/or camera accuracy. We believe our approach could be a very effective baseline for camera control in video diffusion transformers and a good starting point that can be adapted by many follow-up works.

---

### Official Review · Reviewer_qX6r · 2024-11-02

**Soundness:** 3
**Presentation:** 3
**Contribution:** 2
**Rating:** 6
**Confidence:** 5

**Summary:**

This paper provides a method for adding the control of camera movements to video diffusion transformers. The core idea is to represent camera conditions as pixel-level ray condition with Plucker embeddings. A ControlNet inspired module is used to process the camera condition. The model is finetuned on RealEstate10k and is compared to two similar methods both qualitatively and quantitatively.

**Strengths:**

- paper is easy to follow
- the proposed design including the Plücker embedding is reasonable and effective
- comprehensive experiments are conducted and presented in the main manuscript and appendix
- supplemental materials contain video samples to demonstrate the effectiveness

**Weaknesses:**

- The proposed method has been evaluated only on one video diffusion transformer, which raises some concerns on whether its performance can generalize to other pretrained video diffusion transformers.
- I'm curious about the distribution of camera movements evaluated in the experiments, in terms of its diversity and similarity to natural camera movements.
- The novelty is slightly limited, as the task is not new, and ControlNet-like module as well as Plücker embedding have been explored and used before.

minor:
- CameraCtrl was appeared on arXiv in April 2024, which should not be considered as a concurrent work.

**Questions:**

Overall I think this paper provides an effective solution to amend video diffusion transformers with camera control. See weaknesses for questions and discussion.

---

> ### Author Response · Authors · 2024-11-25
>
> We deeply appreciate the reviewer's detailed and positive assessment of our work. Below, we would like to clarify several concerns.
>
> > **Will the method generalize to other transformer-based diffusion models?**
>
> That’s a great and justified question, and to answer it, we implemented our VD3D method on top of a pre-trained text-to-video DiT model in the latent space of the CogVideoX [1] autoencoder. We include these results as part of the revised paper in Table 10 and Table 11. We furthermore include visual samples into the *rebuttal.html*.
>
> Instead of building on top of read attention in FIT, we incorporate the ControlNet conditioning on top of the vanilla attention mechanism of the actual tokens. We provide evaluations in the two Tables below and observe that the vanilla DiT version further improves quality and camera accuracy on out-of-distribution prompts (MSR-VTT). We believe that our proposed method of spatio-temporal Plucker tokens and aligning them with video patch tokens through a ControlNet-type of conditioning mechanism is agnostic to the transformer architecture and will serve as a starting point for follow-up works.
>
> |Method|TransError (RE10K)|RotError (RE10K)|TransError (MSR-VTT)|RotError (MSR-VTT)|
> |:--------|:--------:|:--------:|:--------:|:--------:|
> |MotionCtrl|0.518|0.161|0.627|0.148|
> |CameraCtrl|0.532|0.165|0.578|0.220|
> |Ours (FIT DiT)|**0.409**|**0.043**|0.504|0.050|
> |Ours (vanilla DiT)|0.421|0.056|**0.486**|**0.047**|
>
> |Method|FID (RE10K)|FVD (RE10K)|CLIP (RE10K)|FID (MSR-VTT)|FVD (MSR-VTT)|CLIP (MSR-VTT)|
> |:--------|:--------:|:--------:|:--------:|:--------:|:--------:|:--------:|
> |MotionCtrl|1.50|52.30|0.2708|9.97|183.57|0.2677|
> |CameraCtrl|2.28|66.31|0.2730|8.47|181.90|0.2690|
> |Ours (FIT DiT)|1.40|42.43|0.2807|7.80|165.18|0.2689|
> |Ours (vanilla DiT)|**1.21**|**38.57**|**0.2834**|**6.88**|**137.62**|**0.2790**|
>
> [1] Yang et al., Cogvideox: Text-to-video diffusion models with an expert transformer, arXiv 2024
>
> > **Evaluation on out-of-distribution trajectories.**
>
> In addition to the quantitative evaluations of random out-of-distribution camera trajectories presented in Table 8 of our original submission, we have now conducted a more extensive analysis. The new evaluation tests 11 distinct out-of-distribution trajectories across 1000 different scenes for both RealEstate10K and MSR-VTT, shown in the Table below. We visualize 16 prompts for these trajectories, resulting in 176 new visual results in total in the *rebuttal.html*. This comprehensive analysis is now available in Table 9 of the revised manuscript.
>
> |Method|TransError (RE10K)|RotError (RE10K)|TransError (MSR-VTT)|RotError (MSR-VTT)|
> |:--------|:--------:|:--------:|:--------:|:--------:|
> |MotionCtrl|0.451|0.095|0.456|0.146|
> |CameraCtrl|0.369|0.088|0.479|0.135|
> |Ours|**0.236**|**0.041**|**0.258**|**0.050**|
>
> > **ControlNet and Plucker camera embeddings were explored before.**
>
> Our work addresses fundamentally different challenges from ControlNet in several aspects. First, ControlNet was specifically designed for UNet architectures, while we develop conditioning mechanisms for transformer-based models, which require entirely different approaches due to their distinct computational nature. Second, while ControlNet successfully handles strong spatial signals like edge maps or depth masks that largely determine image structure, we tackle the challenging problem of camera pose conditioning— a much weaker signal that must influence both spatial and temporal aspects of video generation. The integration of such conditioning into joint spatio-temporal transformer computation presents unique technical challenges that require novel solutions.
>
> > **CameraCtrl is not concurrent.**
>
> We respectfully want to clarify that our research, which began in February, developed its approach independently. Moreover, while CameraCtrl is currently under review at ICLR, we have provided comprehensive comparisons with their method throughout our paper. These comparisons demonstrate that their UNet-based approach does not translate effectively to transformer architectures, which required us to develop fundamentally different solutions for camera control in the context of joint spatio-temporal processing.

---

> > ### Comment · Reviewer_qX6r · 2024-11-26
> > **Official Response by Reviewer qX6r**
> >
> > Thanks authors for the detailed response to my questions. I appreciate the additional results for other transformer-based diffusion models and more ood camera trajectories, which shall be included in the revision to make this paper more solid. My concerns have been addressed. While I still think novelty is slightly limited, but this does not affect the value of this paper.

---

### Official Review · Reviewer_uN7u · 2024-11-03

**Soundness:** 2
**Presentation:** 2
**Contribution:** 2
**Rating:** 3
**Confidence:** 5

**Summary:**

The method proposes a controlnet-like architecture for a private video diffusion model by including plucker coordinates as camera control.

**Strengths:**

- The proposed controlnet design outperforms other model variants designed by the authors. The evaluations are thoroughly conducted for the design choices. Detailed ablations are provided.

**Weaknesses:**

- The proposed framework overfits on the trajectories that are seen during training. Though the authors provide quantitative comparisons in Tab. 8, no visual comparisons are provided.
- Though the performance is impressive, the technical contribution is limited in the proposed framework. Training a ControlNet for diffusion transformer is not new, as shown in [1]. Using Plucker coordinates for camera control is not new, as shown in CameraCtrl (He et al., 2024a).

[1] Chen J, Wu Y, Luo S, et al. Pixart-{\delta}: Fast and controllable image generation with latent consistency models[J]. arXiv preprint arXiv:2401.05252, 2024.

**Questions:**

Can the authors please comment on the above-mentioned weaknesses?

**Details Of Ethics Concerns:**

Though the authors are transparent on the fine-tuning data (RealEstate10K), the pre-training data for this proposed framework is unknown, potentially containing copyrighted data. It may also be contaminated with the test sets of the RealEstate10K data or MSR-VTT data, making the reported results in Tab. 2 concerning.

To ensure reproducibility as strongly recommended in the ICLR author guide, the authors are encouraged to adapt the proposed framework to publicly available pre-trained models.

---

> ### Author Response · Authors · 2024-11-25
>
> We thank the reviewer for their thorough feedback. Below, we address the raised concerns.
>
> > **Overfitting on RealEstate10K camera trajectories.**
>
> We respectfully note that all our evaluations were performed on the test-set RealEstate10K camera trajectories. Regarding the out-of-distribution cameras, as the reviewer rightfully pointed out, we provide quantitative results with random camera trajectories in Table 8 in Appendix B. With the current update, we include the results for non-random, user-defined trajectories in the *rebuttal.html* as part of the supplementary, which will be incorporated into *main.html* for the final version. We also provide new quantitative evaluations for these trajectories below and in Table 9 in the revision of the paper. These new trajectories also involve camera movements with significant directional changes including rotations and translations. We observe that our method generalizes to camera trajectories with variable rotations and translations.
>
> |Method|TransError (RE10K)|RotError (RE10K)|TransError (MSR-VTT)|RotError (MSR-VTT)|
> |:--------|:--------:|:--------:|:--------:|:--------:|
> |MotionCtrl|0.451|0.095|0.456|0.146|
> |CameraCtrl|0.369|0.088|0.479|0.135|
> |Ours|**0.236**|**0.041**|**0.258**|**0.050**|
>
> > **Plucker embeddings have already been explored in CameraCtrl.**
>
> We respectfully note that our work developed the use of Plücker coordinates for camera control independently, and our key technical contribution extends beyond just the idea of using them— we developed a novel projection mechanism that effectively integrates camera information into the representation space of video diffusion transformers with joint spatio-temporal layers. This specialized approach is essential, as transformer architectures process information fundamentally differently from UNet-based models and require unique solutions for effective camera control.
>
> > **PixArt-δ has already explored ControlNets for a transformer-based diffusion model.**
>
> We respectfully note that ​while PixArt-δ developed a ControlNet-based mechanism for transformer-based diffusion, it is limited to spatial conditioning with spatial signals. Spatial conditioning signals explored by ControlNet and PixArt-δ (e.g., canny edges, depth/normal/segmentation maps, etc.) are very strong and almost fully describe the structure of the output image. In our paper, we explore conditioning mechanisms for camera poses, which is a *weak* spatio-temporal signal with an intricate temporal component. Moreover, we explore it for transformer-based diffusion models with joint spatio-temporal computation which adds an extra level of complexity because these dimensions are entangled. We thank the reviewer for their valuable pointer and added the relevant discussion in Section 2 of the paper.
>
> > **Details on the pre-training data of the backbone model.**
>
> We appreciate the reviewer's attention to the matter of training data in the backbone model upon which our work is built. However, we want to clarify that we obtained only the upstream model weights from this model, which were published in a paper at CVPR 2024. Our paper focuses entirely on our novel 3D camera control methodology, which we developed on top of this previously established model.
>
> In this case, we believe the 3D camera control methodology should be the focus of the review as is common when new contributions are made on top of existing work. We note that many important papers have been built upon closed models, for instance:
> - DreamFusion (ICLR’23 Best Paper) builds upon Imagen (NeurIPS’22 Best Paper)
> - Cat3D (NeurIPS’24 Oral) builds upon an unnamed upstream LDM model.
> - Magic3D (CVPR’23) builds upon e-Diff-I
> - VidPanos (SIGGRAPH Asia’24) builds upon Lumiere
>
> To ensure consistency in the criteria used to evaluate our contribution, we would appreciate a similar approach to be taken with the review of our paper.
>
> > **Pre-training data can be potentially contaminated with test data.**
>
> As to the potential contamination of pre-training data with test data, we confirm that the pre-training data does not overlap with neither train nor test sets of RealEstate10K or MSR-VTT, ensuring the validity of the results reported in Table 2.
>
> > **Reproducibility concern.**
>
> In our original submission, we provided a detailed description of our methodology in Section 3, comprehensive experimental details in Section 4, and complete training and architectural specifications in Appendix C. This constitutes a solid foundation for future projects to reproduce and build upon our work.
>
> To further enhance reproducibility, we include the source code of our camera-controlled FIT block — the key component of our method — as supplementary material. Additionally, we are committed to providing any technical details that reviewers find necessary for complete reproduction of our results.
>
> We have added a Reproducibility Statement as Section 6, summarizing our commitment to reproducibility and the available resources.

---

> > ### Comment · Reviewer_uN7u · 2024-11-28
> >
> > Thanks to the authors for the detailed response. I have some follow-up questions:
> >
> > - Can the authors please clarify what exactly is the technical contribution of the paper? I carefully read the other reviews and all reviewers share the same concern that the novelty of this work is very limited.
> >    - For example, what is `we developed a novel projection mechanism`? How does that differ from a controlnet-like structure? If I understand correctly, the difference is that one shouldn't use multiple layers but only one layer for projection.
> >    - I understand that PixArt-δ is not designed for temporal conditioning signals, but what is the difference from their design? The authors keep suggesting there is `an extra level of complexity`, but the proposed approach is simply concatenating along the extra dimension.
> > - I share the concern of Reviewer ZZhZ that purely evaluating on SnapVideo is sketchy.
> >     - While I appreciate the authors' effort in providing a vanilla DiT version, I get confused since the performance of the vanilla DiT version is superior. In this case, what's the purpose of designing VD3D on top of SnapVideo? If we switch the architecture to a vanilla DiT, what is the novelty then? Do we still have a `novel projection mechanism`, or has the `novel projection mechanism` now become a vanilla controlnet block? My question comes from L311 and L1185 that the `novel projection` is operating on the FiT's `read` layer.
> >     - Are the baselines (MotionCtrl and CameraCtrl) also implemented using a vanilla DiT for the comparison? Are the vanilla DiT versions also trained on the same pre-training data for SnapVideo?
> >     - How is the CameraCtrl baseline produced? On L381 it says, `For CameraCtrl, we fine-tune the original camera encoder module and use this to produce the latent vectors in the SnapVideo model.` Are the authors fine-tuning a camera encoder trained from AnimateDiff? I might have missed this, but what happens when properly implementing a SnapVideo version of CameraCtrl by using zero-convolution plus a copy of SnapVideo layers? Are all CameraCtrl variants mentioned in the paper implemented the same way?
> > - Can the authors please comment on Table 8? When evaluating videos without translation, does Particle-SfM produce reliable results?

---

> > > ### Author Response · Authors · 2024-11-28
> > >
> > > Thanks, we want to clarify the remaining concerns.
> > >
> > > > **Technical contributions of the paper**
> > >
> > > Our main technical contribution is proposing how to incorporate camera control into transformer-based video generators. After extensive experimentation and analysis, which we include throughout the paper with many ablations, we came to our final setup. First, we propose to align camera input and video patch tokens spatiotemporally. For this, we represent extrinsics and intrinsics as spatio-temporal Plucker embeddings, patchify them into spatio-temporal Plucker tokens, and then align them with the video tokens. This procedure is unique to transformer-based models and was not investigated by any other work, as previous works only investigated U-Net-based video models. Second, we propose to align these spatiotemporal Plucker tokens with the video tokens using a ControlNet-inspired setup. Other works, including MotionCtrl and CameraCtrl, do not use ControlNet to infuse cameras into the base video model. We observe that spatio-temporals ControlNets are very effective and degrade little motion and quality, as shown in the experiments. Note that using ControlNet is only possible after representing cameras in the same spatio-temporal patch space as the original video tokens as part of the first step we are doing. Novelty is not always about inventing a new complicated layer or mechanism, but it can also be novel connecting different mechanisms and showing that a simple approach is more effective. In fact, the community typically prefers to build upon simple approaches that just work. We are not claiming to invent Plucker or ControlNet; it’s the technical solution of representing cameras as spatiotemporal Plucker tokens, using a ControlNet-type of setup to align them spatiotemporal video tokens pixel-wise, and demonstrating that this technique is significantly more effective than any other proposed technique. We also demonstrate with extensive experiments that using raw extrinsics (MotionCtrl) or camera encoders without ControlNet-type of camera alignment (CameraCtrl) work significantly worse, degrading quality, motion, and/or camera accuracy. We believe our approach could be a very effective baseline for camera control in video diffusion transformers and a good starting point that can be adapted by many follow-up works.
> > >
> > > > **Comparisons based on FIT-based SnapVideo**
> > >
> > > At the time of submission, no vanilla video DiT model was available; hence, we used the FIT-based model. While one part of the analysis is incorporating the camera in the read-attention process, this is not our main novelty. With our new experiment, we want to highlight that our approach applies to different transformer-based architectures. As outlined above, our main technical novelty lies in representing cameras as patchified spatiotemporal Plucker tokens and second incorporating them spatiotemporally with a spatiotemporal ControlNet into the spatiotemporal video tokens. This has not been done before, as previous works neither use transformer-based models nor a ControlNet setup to represent and align cameras with video features.
> > >
> > > > **Vanilla DiT baseline**
> > >
> > > We are currently training a MotionCtrl and CameraCtrl variant on top of the same vanilla DiT model and will include those results when they are ready at the end of the rebuttal period. The intention of providing results of our method on top of the Vanilla DiT baseline for the rebuttal was to show that our method generalizes to different transformer-based architectures. We observe that the camera accuracy for FIT is very similar to the DiT version, which is mainly visible in the rotation errors. Hence, the vanilla DiT model does not change the results much regarding camera accuracy. This highlights that our model works similarly precisely independent of FIT or vanilla DiT. Note that MotionCtrl and CameraCtrl heavily struggle with large rotation errors, i.e., the camera does not point toward the correction direction. The intention of the rebuttal experiment for vanilla DiT was not to conduct another thorough analysis on another backbone, as this computationally is very expensive. It is common for camera control and multi-view diffusion works to conduct the analysis and comparisons on a single video model. We have already made sure by using the same base model throughout the paper that we do not have any advantage over AnimateDiff-based approaches. We also included visual results in the *rebuttal.html* for the vanilla DiT model, showing high-quality, high-motion videos with camera control. The main point of this experiment is to show that our method does not converge to static scenes for another backbone, as commonly observed in other camera control works when fine-tuning on RealEstate10K. Furthermore, we demonstrate that the vanilla DiT results in the *rebuttal.html* are precise. Hence, our proposed mechanism generalizes and can be a starting point for many transformer-based follow-up works.

---

> > > > ### Author Response · Authors · 2024-11-28
> > > >
> > > > > **CameraCtrl implementation**
> > > >
> > > > We believe there is a misunderstanding regarding the implementation of CameraCtrl. We do **not** fine-tune the pre-trained camera encoder from AnimateDiff since this would not generalize well, and even zero-convolutions might not easily adapt to the SnapVideo model. What we mean by “original camera encoder module” is the code and the architecture that we adapt into the same base model codebase we use, i.e., we do not use the weights from AnimateDiff-trained camera encoders. We start fine-tuning the camera encoder module from the **same** SnapVideo model, using the **same** batch size, **same** number of iterations, and **same** parameter size as our proposed method for fair comparisons. We will update the draft to emphasize this as part of the next revision of the paper.
> > > >
> > > > > **Particle-SfM reliability**
> > > >
> > > > Particle-SfM, and generally SfM methods, are not perfect and cannot always estimate the camera poses in a scene correctly. Sometimes, the SfM process will not converge, and we follow the common practice of repeating the SfM process until it converges to a solution. Unfortunately, no better camera pose estimators are available for in-the-wild videos, and other works, such as CameraCtrl, use the same setup as us to evaluate camera accuracy.
> > > >
> > > > We include 176 visualizations in the *rebuttal.html* in the supplementary file to provide more evidence for our performance on out-of-distribution camera trajectories. We urge the reviewer to assess them on their own: VD3D has very precise camera accuracy for diverse, user-defined camera trajectories, like translations, rotations, and their combinations. We believe those results are very reliable and address all the concerns regarding overfitting to RealEstate10K trajectories.

---

> > > > > ### Comment · Reviewer_uN7u · 2024-11-29
> > > > >
> > > > > Thanks for the response. Can the authors please clarify how exactly CameraCtrl is re-implemented? I understand the backbone SnapVideo is the same. Please provide as much information as possible because the current descriptions seem problematic.

---

> > > > > > ### Author Response · Authors · 2024-11-29
> > > > > >
> > > > > > Thank you for the follow-up questions, we are happy to provide more details on the CameraCtrl baseline to clarify any remaining concerns.
> > > > > >
> > > > > > For CameraCtrl, we use the original code and integrate this code into the same base transformer model we use. Specifically, extrinsics and intrinsics are encoded using Plucker embeddings. Subsequently, the Plucker embeddings are encoded with the original camera encoder architecture that uses 2D ResNet and temporal attention layers. Note that the original camera encoder uses a pooling operation to downsample the Plucker embeddings to the different downsampled feature resolutions in the U-Net architecture. Since we use a transformer with constant dimensions across blocks, we remove the downsampling operations and keep the spatial dimensions constant. Otherwise, we follow the original CameraCtrl encoding setup and use the same patchification process as we do to align the embeddings with the spatio–temporal patches of the base model. The features are injected into the respective transformer blocks with a sum and linear layer. Due to zero-initialization, the camera is smoothly integrated into the main model. We initialize all layers with the same initialization scheme as the original CameraCtrl work and freeze the base model. While the original U-Net-based CameraCtrl integrates the features into the temporal attention layer, there are no separate temporal layers in spatio-temporal transformers that jointly model image and time dimensions. Hence, we extensively experimented with where to inject the features similarly as we experimented with our approach. In the end, the best approach was to map the output of the linear layer into the latent tokens, where self-attention instead of temporal attention layers are applied to these latent tokens in the transformer block. One key difference is that we use a ControlNet-based approach that performs attention between video tokens and Plucker tokens before integrating them into the base model, which is crucial.
> > > > > >
> > > > > > Furthermore, we have extra comparisons in the supplementary, showing alternative implementation strategies for MotionCtrl and CameraCtrl. The main comparisons use the variants that performed the best. Throughout our comparison process, we tried several ways to incorporate CameraCtrl into the transformer-based architecture. This also led to the certainty that our ControlNet-inspired approach is the most effective way to control camera in video diffusion transformers, especially w.r.t. preserving motion quality and visual quality. If there is any other way that the reviewer thinks we could explore or any concrete problem, we are happy to conduct an experiment on that. But we believe that we have already put a lot of effort into these comparisons, spending a lot of compute on training different variants to find the best CameraCtrl version based on our base model.
> > > > > >
> > > > > > Importantly, before conducting any comparisons with CameraCtrl, **we were in contact with the CameraCtrl authors to make absolutely sure that we conducted fair comparisons** by getting their full code with evaluation scripts, as the public code was missing parts of the evaluation code. Consequently, **we are absolutely certain that our implementation and comparisons with CameraCtrl are correct and involve significant efforts to ensure fairness and comparability**.
> > > > > >
> > > > > > We will add these extra clarifications to the paper's revision. Thank you for the suggestions to further clarify the CameraCtrl baseline implementation. If any unaddressed concerns are left, we would be grateful to the reviewer for pointing them out since the discussion phase is closing soon.

---

> > > > > > > ### Author Response · Authors · 2024-12-02
> > > > > > >
> > > > > > > > **Further vanilla DiT results**
> > > > > > >
> > > > > > > As promised, we trained MotionCtrl and CameraCtrl on the same additional vanilla DiT baseline. We observe the same benefits as with our originally used transformer backbone and show generalization of our method across architectures. We will add these results to the next revision of the paper, as it is currently not possible to update the paper on openreview.
> > > > > > >
> > > > > > > |Method|TransError (RE10K)|RotError (RE10K)|TransError (MSR-VTT)|RotError (MSR-VTT)|
> > > > > > > |:--------|:--------:|:--------:|:--------:|:--------:|
> > > > > > > |MotionCtrl (vanilla DiT)|0.501|0.145|0.602|0.152|
> > > > > > > |CameraCtrl (vanilla DiT)|0.513|0.138|0.559|0.195|
> > > > > > > |Ours (vanilla DiT)|**0.421**|**0.056**|**0.486**|**0.047**|
> > > > > > >
> > > > > > > |Method|FID (RE10K)|FVD (RE10K)|CLIP (RE10K)|FID (MSR-VTT)|FVD (MSR-VTT)|CLIP (MSR-VTT)|
> > > > > > > |:--------|:--------:|:--------:|:--------:|:--------:|:--------:|:--------:|
> > > > > > > |MotionCtrl (vanilla DiT)|1.37|44.62|0.2752|9.68|157.90|0.2684|
> > > > > > > |CameraCtrl (vanilla DiT)|2.13|53.72|0.2748|8.32|152.88|0.2723|
> > > > > > > |Ours (vanilla DiT)|**1.21**|**38.57**|**0.2834**|**6.88**|**137.62**|**0.2790**|
> > > > > > >
> > > > > > > > **Rebuttal Period Summary**
> > > > > > >
> > > > > > > Thank you again for your detailed feedback and engagement in the review process. We greatly appreciate your time and effort in evaluating our work. Throughout this review process, we have carefully and thoroughly addressed all the concerns raised in all your feedback posts across three discussion threads, providing new evaluations, experimental results, clarifications, and explanations. All this valuable feedback will be incorporated into the final version of our work.
> > > > > > > With the discussion thread closing today, we wanted to see if there might be any other issues we could address to help improve the paper. Given the thorough discussion and our efforts to address all issues comprehensively, we would greatly appreciate it if you could consider revising your score to reflect the current state of the paper.
> > > > > > > Thank you again for your time and thoughtful feedback throughout this review process.

---

> > > > > > > > ### Comment · Reviewer_uN7u · 2024-12-03
> > > > > > > >
> > > > > > > > Thank you for your response. I have a few follow-up questions regarding the experiments and noticed that some points from my previous concerns may not have been addressed. For clarity, I’ve compiled all the questions below:
> > > > > > > >
> > > > > > > > - How did the CameraCtrl authors verify that you implemented CameraCtrl correctly? Did they read your implementation?
> > > > > > > > - What is the trainable parameter size for the re-implementation of CameraCtrl?
> > > > > > > > - What is the architecture for the camera encoder for CameraCtrl, when experimenting with SnapVideo and the DiT?
> > > > > > > > - How many training iterations are conducted for the CameraCtrl and MotionCtrl versions?
> > > > > > > > - What is the vanilla DiT you are referring to? Is it the pre-trained CogVideoX? What is the parameter size of this new backbone? What dataset was it pre-trained on?
> > > > > > > > - Is my understanding correct that the CameraCtrl version is using a sum and linear layer, while VD3D replaces the sum and linear layer with attention layers?
> > > > > > > > - Does VD3D have a camera encoder module? If not, how are the trainable parameters maintained the same for VD3D and CameraCtrl?

---

> > > > > > > > > ### Author Response · Authors · 2024-12-04
> > > > > > > > >
> > > > > > > > > Thank you for your further questions, we are happy to address all remaining concerns.
> > > > > > > > >
> > > > > > > > > > **How did the CameraCtrl authors verify that you implemented CameraCtrl correctly? Did they read your implementation?**
> > > > > > > > >
> > > > > > > > > As specified in our previous response, *“we were in contact with the CameraCtrl authors to ensure fair comparisons by obtaining their full code including the evaluation scripts.”* However, it would not be reasonable to expect the CameraCtrl authors to review our code to verify its correctness. That said, there should be no concern in this regard, as we used the original CameraCtrl code directly without making any modifications when integrating it into our codebase.
> > > > > > > > >
> > > > > > > > > > **What is the trainable parameter size for the re-implementation of CameraCtrl?**
> > > > > > > > >
> > > > > > > > > 230M trainable parameters, the same as for our method VD3D.
> > > > > > > > >
> > > > > > > > > > **What is the architecture for the camera encoder for CameraCtrl, when experimenting with SnapVideo and the DiT?**
> > > > > > > > >
> > > > > > > > > The camera encoder is exactly the same as the one proposed in the original CameraCtrl work, and we use that code and architecture without modifications for both the SnapVideo and vanilla DiT backbone experiments. We only adjust the dimensions to be compatible with the base models we use, as the U-Net used in the original CameraCtrl work has dimensions different from the transformer-based models we use. You can see it as a vanilla adaptation of the CameraCtrl work, originally developed for U-Net architectures, to transformer-based architectures.
> > > > > > > > >
> > > > > > > > > > **How many training iterations are conducted for the CameraCtrl and MotionCtrl versions?**
> > > > > > > > >
> > > > > > > > > The same number of training iterations as for our method, i.e., 50,000 iterations. We will revise Sec. C in the appendix to emphasize that CameraCtrl and MotionCtrl were also trained with 50,000 iterations.

---

> > > > > > > > > > ### Author Response · Authors · 2024-12-04
> > > > > > > > > >
> > > > > > > > > > > **What is the vanilla DiT you are referring to? Is it the pre-trained CogVideoX? What is the parameter size of this new backbone? What dataset was it pre-trained on?**
> > > > > > > > > >
> > > > > > > > > > We used a vanilla DiT base model for additional experiments demonstrating generalizability across different transformer-based architectures. Concretely, we used an internally available video diffusion transformer with vanilla DiT architecture with 11.5B parameters in total. While we provide its detailed information below, we emphasize that it is not central to the contributions of our work since (i) the DiT design is basically the same for all modern video DiT models (CogVideoX, Sora, MovieGen, Mochi-1, Allegro, OpenSora, LuminaT2X, etc.), (ii) the DiT design is **orthogonal** to our proposed method on camera control, and (iii) all the baselines have been trained on top of **exactly the same DiT model**. The key reason why we used our internally available model instead of CogVideoX or some other open-source one is that it is paired with our existing codebase and allowed us to quickly re-implement the camera control method and the baselines on very short notice for this rebuttal. It is important to note that it is already taking a lot of time and computational effort to integrate CameraCtrl and MotionCtrl into our base model. Now, we show comparisons with (i) the original MotionCtrl and CameraCtrl, (ii) MotionCtrl and CameraCtrl integrated into our originally used FIT-based transformer architecture, and (iii) MotionCtrl and CameraCtrl integrated into an additional vanilla DiT architecture operating in the latent space of the CogVideoX autoencoder. This is a significant effort and significantly more than what previous works do.
> > > > > > > > > >
> > > > > > > > > > **Base video DiT architecture details.**
> > > > > > > > > > The video DiT architecture follows the design of other contemporary video DiT models (e.g., Sora, MovieGen, OpenSora, LuminaT2X, and CogVideoX). As the backbone, it incorporates a transformer-based architecture with 32 DiT blocks. Each DiT block includes a cross-attention layer for processing text embeddings (produced by the T5-11B model), a self-attention layer, and a fully connected network with a ×4 dimensionality expansion. Attention layers consist of 32 heads with RMSNorm for query and key normalization. Positional information is encoded using 3D RoPE attention, where the temporal, vertical, and horizontal axes are allocated fixed dimensionality within each attention head (using a 2:1:1 ratio). LayerNorm is applied to normalize activations within each DiT block. A pre-trained CogVideoX autoencoder is utilized for video dimensionality reduction, employing causal 3D convolution with a 4×8×8 compression rate and 16 channels per latent token. The model features a hidden dimensionality of 4,096 and comprises 11.5B parameters. It leverages block modulations to condition the video backbone on rectified flow timestep information, SiLU activations, and 2×2 ViT-like patchification of input latents to reduce sequence size.
> > > > > > > > > >
> > > > > > > > > > **Base video DiT training details.**
> > > > > > > > > > The base DiT model is optimized using AdamW, with a learning rate of 0.0001 and weight decay of 0.01. It is trained for 750,000 iterations with a cosine learning rate scheduler in bfloat16. Image animation support is incorporated by encoding the first frame with the CogVideoX encoder, adding random Gaussian noise (sampled independently from the video noise levels), projecting via a separate learnable ViT-like patchification layer, repeating sequence-wise to match video length, and summing with the video tokens. Training incorporates loss normalization and is conducted jointly on images and videos with variable resolutions ($256$, $512$, and $1024$), aspect ratios ($16:9$ and $9:16$ for videos; $16:9$, $9:16$, and $1:1$ for images), and video lengths (ranging from 17 to 385 frames). Here we emphasize again that the pre-training data does not overlap with neither train nor test sets of RealEstate10K or MSR-VTT. Videos are generated at 24 frames per second, and variable-FPS training is avoided due to observed performance decreases for target framerates without fine-tuning.
> > > > > > > > > >
> > > > > > > > > > **Base video DiT inference details.**
> > > > > > > > > > Inference uses standard rectified flow without stochasticity. We find forty steps to balance quality and sampling speed effectively. For higher resolutions and longer video generation, a time-shifting strategy similar to Lumina-T2X is used, with a time shift of $\sqrt{32}$ for $1024$-resolution videos. We will include these details in our revisions. We want to emphasize that these are the details of the **additional transformer model** we used for the rebuttal. This was mainly done to show the generalization capabilities of our approach to another transformer-based architecture. Our main experiments in the original paper submission already include complete descriptions of the base model SnapVideo published in CVPR 2024.

---

> > > > > > > > > > > ### Author Response · Authors · 2024-12-04
> > > > > > > > > > >
> > > > > > > > > > > > **Is my understanding correct that the CameraCtrl version is using a sum and linear layer, while VD3D replaces the sum and linear layer with attention layers?**
> > > > > > > > > > >
> > > > > > > > > > > Your understanding of CameraCtrl is correct. For VD3D, we replace the sum and linear layer mechanism with a ControlNet-type block described in the method section (Sec. 3). Hence, we use linear layers and attention blocks with sums inside the ControlNet block. We kindly refer to Fig. 3 in the paper, where this is visualized.
> > > > > > > > > > >
> > > > > > > > > > > > **Does VD3D have a camera encoder module? If not, how are the trainable parameters maintained the same for VD3D and CameraCtrl?**
> > > > > > > > > > >
> > > > > > > > > > > VD3D does not have a camera encoder module. We adjust the hidden dimensions of each method to accumulate the same number of total parameters across methods. This also leads to the same memory consumption and the same batch size across the methods for fair comparisons. We previously experimented with the original parameter count of CameraCtrl and observed similar results to those of the adjusted parameter count.

---

### Official Review · Reviewer_B4pY · 2024-11-03

**Soundness:** 3
**Presentation:** 3
**Contribution:** 2
**Rating:** 8
**Confidence:** 4

**Summary:**

This paper presents a camera control method for transformer-based video generation models that enhances control while ensuring visual quality. The proposed approach aligns the video perspective with predefined camera trajectories, improving controllability. The authors claim this is the first study to employ ControlNet-like guidance for global spatiotemporal transformer video generation models, in contrast to the more commonly used U-Net architecture. Moreover, the evaluation demonstrates that both the video quality and adherence to the input camera trajectories are state-of-the-art.

**Strengths:**

Camera control during the video generation process is a significant issue. As more foundational models adopt transformer architectures, exploring control mechanisms for these models becomes crucial. This paper is the first to investigate how to better utilize camera trajectory parameters for transformer-based video generation models, using SnapVideo as the foundational model. The design is well thought out, and the evaluation is rigorous. The strengths of the paper are as follows:

- Unlike the spatiotemporal decoupling generation of U-Net structures, transformer-based video generation considers spatiotemporal video tokens globally, which means it cannot directly leverage the advantages of spatiotemporal decoupling. This paper overcomes this limitation by being the first to explore control specifically for spatiotemporal transformers. This shift in foundational model structure is critical and provides a solid engineering foundation for future work.

- The authors use Plücker embeddings to convert camera intrinsic and extrinsic parameters into pixel-level controls, which match the shape of video tokens. This information is then introduced through read cross-attention layers. While this approach is a straightforward combination of existing methods, it has been validated as effective for transformer-based video generation models, providing valuable experimental insights.

- The paper includes comprehensive evaluations and ablation studies, conducting both qualitative and quantitative experiments regarding video content quality and camera control, with well-defined criteria. The evaluation of baseline models is fair, making the transition to the new foundational model structure more convincing.

**Weaknesses:**

- While camera control is the central problem addressed in this paper, the camera trajectories used primarily come from the RealEstate10K dataset, which, as observed in the visual results, mostly follow smooth, straight lines. There is a lack of consideration and experimentation with trajectories of varying difficulty, such as those involving significant directional changes. This raises some questions regarding the trajectory settings.

- There have been several prior works in the 3D multi-view generation field that focus on similar camera control issues, such as the referenced *Cat3D*, which also employed Plücker embeddings and attention calculations. The distinction between spatiotemporal decoupling and other network characteristics is a design feature intrinsic to the architecture. In exploring DiT-based generation, there have also been multiple studies investigating spatiotemporal decoupling, such as *Latte: Latent Diffusion Transformer for Video Generation*. Therefore, the novelty of this work lies more in applying existing designs to spatiotemporal transformers rather than presenting a technological innovation. However, the state-of-the-art results under the new configuration indeed serve as an important engineering reference for future directions.

**Questions:**

- Have there been additional relevant experiments regarding camera trajectories, such as comparisons of control quality for generated trajectories of varying complexity?

At this time, I have no further questions.

---

> ### Author Response · Authors · 2024-11-25
>
> We deeply appreciate the reviewer's detailed and positive assessment of our work. Below, we would like to clarify several concerns.
>
> > **Evaluation on diverse camera trajectories**
>
> We respectfully note that we provide quantitative evaluation for random (i.e., not RealEstate10K) camera trajectories in Table 8 in the appendix of the original submission. With the current update, we include 176 new visual results for non-random, user-defined camera trajectories in the rebuttal.html as part of the supplementary, which will be incorporated into main.html for the final version. We also provide new quantitative evaluations for these trajectories below and in Tab. 9 in the revision of the paper. These new trajectories also involve camera movements with significant directional changes including both rotations and translations, as suggested by the reviewer. We observe that our method generalizes to input camera trajectories with variable rotations and translations.
>
> |Method|TransError (RE10K)|RotError (RE10K)|TransError (MSR-VTT)|RotError (MSR-VTT)|
> |:--------|:--------:|:--------:|:--------:|:--------:|
> |MotionCtrl|0.451|0.095|0.456|0.146|
> |CameraCtrl|0.369|0.088|0.479|0.135|
> |Ours|**0.236**|**0.041**|**0.258**|**0.050**|
>
> > **Novelty concern**
>
> We demonstrated in Table 2 and through qualitative results in the supplementary materials that existing ControlNet-like architectures do not directly translate to diffusion transformers with joint spatiotemporal processing. Extensive experimentation was required to develop a setup that achieves precise camera control while maintaining motion quality and visual fidelity. Importantly, Latte's work is orthogonal to ours—while they focus on developing a diffusion transformer backbone, our contribution lies in creating camera conditioning mechanisms for such models and addressing the inherent challenges that arise. We appreciate this relevant reference and have incorporated Latte into our related work discussion in the revised paper.

---

> > ### Comment · Reviewer_B4pY · 2024-11-26
> >
> > I have no more questions. Precise control of camera trajectory is an important feature of future video-generation products. Thanks to the author for sharing more technical details for reference when reproducing. I'll keep my current score.

---

> > > ### Comment · Reviewer_uN7u · 2024-11-28
> > >
> > > Dear Reviewer B4pY,
> > >
> > > Thanks for the detailed comments and for sharing your opinions. I agree with you that this work is applying existing designs to spatiotemporal transformers rather than presenting a technological innovation. However, I find it hard to understand the contributions of the proposed method.
> > >
> > > While I agree that the proposed method outperforms vanilla MotionCtrl and CameraCtrl, I would argue that this performance gain mainly comes from SnapVideo versus AnimateDiff, where SnapVideo enjoys better pre-training data and larger parameter size. Regarding the experiments using SnapVideo, I find it confusing about their re-implementation of CameraCtrl. According to L381, the authors integrate a camera encoder from AnimateDiff with SnapVideo. While this could work, I don't think it is the correct way to implement CameraCtrl for SnapVideo, where zero-convolutions and a correct weight copy should be necessary. Therefore, I remain confused about the new configuration proposed by this work.
> > >
> > > Please feel free to share your thoughts.

---

> > > > ### Author Response · Authors · 2024-11-28
> > > >
> > > > Dear Reviewer uN7u,
> > > >
> > > > We appreciate your interest in engaging with Reviewer B4pY's assessment of our work. However, as the authors, we would like to address the comparison concerns directly, which we also outlined in response to Reviewer uN7u.
> > > >
> > > > > **CameraCtrl implementation**
> > > >
> > > > We believe there is a misunderstanding regarding the implementation of CameraCtrl. We do **not** fine-tune the pre-trained camera encoder from AnimateDiff since this would not generalize well, and even zero-convolutions might not easily adapt to the SnapVideo model. What we mean by “original camera encoder module” is the code and the architecture that we adapt into the same base model codebase we use, i.e., we do not use the weights from AnimateDiff-trained camera encoders. We start fine-tuning the camera encoder module from the **same** SnapVideo model, using the **same** batch size, **same** number of iterations, and **same** parameter size as our proposed method for fair comparisons. We will update the draft to emphasize this as part of the next revision of the paper.

---

### Author Response · Authors · 2024-11-25
**Rebuttal**

We deeply appreciate the reviewers’ thorough feedback and their positive comments regarding our contributions. We addressed all the raised concerns via targeted responses for each feedback. We summarize our updates here:

- We ran the evaluations for our method on out-of-distribution camera trajectories (various translations, rotations, zoomings and their combinations) and included 176 new qualitative results in *rebuttal.html* as part of the updated supplementary file.
- We updated Figure 3 with the method architecture by enhancing its style and improving readability, simplified the notation in Section 3, and connected it better with the notations and paired the blocks in diagrams with exact variables
- We included the reproducibility statement as Section 6 (as per discussion with Reviewer uN7u). To enhance the reproducibility further, we also provided the source code of our camera-conditioned transformer block — the key ingredient of our method. We are willing to include other details if the reviewers would find necessary.
- We implemented our method on top of a vanilla video DiT architecture, ran the corresponding experiments and evaluations. We included the results as Table 10 and Table 11 in the appendix.
- We added several useful references into the discussion of the related work. We thank the reviewers for valuable pointers.

We are eager to engage in further discussions and would love to improve our work with the further suggestions of the reviewers.

---

### Author Response · Authors · 2024-12-04

We are thankful to all the reviewers for their insightful comments. We believe that the additional clarifications and experiments have improved the paper, we worked very hard in the rebuttal period (11 tables in the paper in total now), and we were able to resolve the majority of your concerns. However, reviewer uN7u has been actively trying to convince others with information that we do not believe is fair.

- **Data:**
We reiterate once more that we only used the RealEstate10K *train* split for training camera control. The RealEstate10K and MSR-VTT test datasets are not seen during camera control training. The base model did not see any of these datasets at all during training. Moreover, the base model is shared for all methods used in our comparisons. No method draws advantage from training on different data.

- **Baselines:**
While many concurrent works only compare against the “as-is” implementation from the original codebases, we made the additional effort of implementing U-Net-based approaches into transformer-based models to ensure more direct and fair comparisons. This allows us to eliminate any differences in results stemming from base model differences, base model data used, number of fine-tuning iterations, batch size, parameter size, or computational resources. We aligned all experiments to match these aspects to ensure fair comparisons, leading to comparisons with the original U-Net-based approach and two transformer-based implementations based on the original code bases of MotionCtrl and CameraCtrl.

- **"[...] verify that you implemented CameraCtrl correctly?"**
When integrating it into our codebase, we used the original CameraCtrl code directly, without making any modifications. Furthermore, we reached out to the CameraCtrl authors to obtain the complete evaluation scripts, as those had not been publicly released. Asking a competing group to “peer-review” your code is not common practice in the community. There is a point when one needs to trust the authors that they have done everything in their power to ensure a correct implementation and a fair evaluation. We believe that the quality of results produced by our model are indicative of our ability to implement a model correctly.

---

### Meta-Review · Area_Chair_MFCV · 2024-12-19

**Metareview:**

This paper proposes a camera-control video diffusion model. The central claimed contribution is its transformer-based architecture. The method builds on SnapVideo and introduces a new attn-based conditional module for injecting the camera information. STOA performances are obtained compared to the existing approaches.

Reviewers highlighted the key advantages of the method, particularly the effectiveness of its transformer-based architecture and Plücker embedding conditioning, as evidenced by its strong experimental results and qualitative performance.

A common concern among reviewers was the limited novelty, noting that the main contribution involved replacing the linear layer in the condition block with attention layers, alongside some necessary design adjustments for Plücker embedding injection—an approach that also has been previously explored in camera-controllable video generation.

The AC acknowledged the solid contribution of the work, as validated by its experimental results, and recognized its value in advancing the field.

**Additional Comments On Reviewer Discussion:**

Reviewer `uN7u` had raised several concerns centering around the fairness of comparison with CameraCtrl vs. VD3D. While there are long discussions on the details of the CameraCtrl re-implementation and whether the setting is fair when compared, Reviewer `uN7u` had emphasized three key points: (a) limited novelty, as also noted by other reviewers; (b) potential reproducibility issues due to SnapVideo being private; and (c) likely an unfair comparison. Among these, (c) poses a significant concern that could diminish the paper's contributions.

The ACs carefully reviewed the clarifications provided by the authors. The authors clarified that the trainable parameters were kept consistent, with CameraCtrl utilizing an encoder while VD3D does not, albeit with potentially larger attention modules on the conditional side. The ACs acknowledge that this comparison setup may be reasonable but strongly urge the authors to provide a more comprehensive description of the CameraCtrl re-implementation. Additionally, the authors are encouraged to include further ablations, such as ensuring comparable sizes of the conditional network, to enhance transparency and strengthen the validity of the comparisons.

---

### Decision · Program_Chairs · 2025-01-22

Accept (Poster)